# Active power filter control strategy based on repetitive quasi-proportional resonant control with linear active disturbance rejection control

**Yifei Gao**[1], **Liancheng Zhu**[1*], **Xiaoyang Wang**[1], **Hongshi Wei**[2], **Xiaoguo Lv**[2]

**1** School of Electrical Engineering, Liaoning University of Technology, Jinzhou, China, **2** Research and Development Department, Liaoning Rongxin Xingye Electric Power Technology Company, Limited, Anshan, China

* dq_lczhu@lnut.edu.cn (LZ)

**Data availability statement:** All relevant data are within the manuscript.

## Abstract

Active power filter (APF) is a new type of harmonic mitigation device, and its harmonic mitigation capability mainly depends on the control strategies of current inner loop and voltage outer loop. Traditional proportional integral (PI) control methods cannot track harmonic currents without steady-state error, thereby it leads to the poor tracking accuracy of the harmonic current and is difficult to achieve high-performance harmonic compensation. Thus, in this paper, a repetitive quasi-proportional resonant (QPR) with linear active disturbance rejection control (LADRC) is proposed. Firstly, QPR control is introduced to replace PI control. QPR control can eliminate steady-state error and realize no static error tracking of harmonic current. Meanwhile, QPR control can coordinate control of various frequency components and enhance APF's ability to suppress specific harmonics. Then, the introduction of repetitive control optimizes the QPR control, so that the dynamic performance of the system does not change while further improving the tracking accuracy of the APF system for harmonic current. Finally, LADRC control is used to observe and compensate for the coupled parts of the system. Thereby, it can achieve high-performance decoupling without adding additional sensors. In addition, in order to ensure the stability of the DC-side voltage and enhance its dynamic performance, this paper also designs a voltage outer loop controlled by QPR. The proposed method is verified by real-time digital simulation system (RTDS). The simulation results show that compared with the double closed-loop PI control, the repetitive quasi-proportional resonant with linear active disturbance rejection control (repetitive QPR-LADRC) and QPR double closed-loop control proposed in this paper can not only increase the harmonic current tracking accuracy of APF system, but also improve the dynamic performance of APF system and significantly enhance the harmonic suppression ability of APF.

**Funding:** This work is financially supported by the <2024 Fundamental Research Project (No. LJ212410154019) of the Educational Department of Liaoning Province>, China. We declare that we have no any other financial and personal relationships with other people or organizations that can inappropriately influence our work, there is no professional or other personal interest of any nature or kind in any product, service and/or company that could be construed as influencing the position presented in, or the review of, the revised manuscript entitled "Active power filter control strategy based on repetitive quasi-proportional resonant control with linear active disturbance rejection control".

**Competing interests:** The authors have declared that no competing interests exist.

## Introduction

With the rapid development of modern social science and technology, a large number of non-linear power electronic devices have been applied, and it results in a large number of harmonics and reactive power being injected into the public power grid. Excessive harmonic currents can cause damage to power electronic devices and even lead to safety accidents, then it will seriously affect the safe and stable operation of the public grid [1,2]. Currently, the mainstream harmonic suppression devices are passive filter (PF) and active power filter (APF). Passive filters are simple in structure and are usually composed of capacitive filters (C) and inductive filters (L) [3]. There are three common types of PF: single tuning, double tuning and triple tuning, which provide low impedance branches in parallel with nonlinear loads such as electric traction locomotives, rectifiers and inverters to achieve filtering [4]. In addition, there is a high-pass damping filter, which can provide a low impedance branch for high-order harmonics to achieve high-order harmonic filtering. Therefore, the combination of a single-tuned filter with 5th, 7th, 11th and 13th harmonic frequencies and a second-order passive high-pass filter is often used in engineering to eliminate AC harmonics [5]. However, PF can only suppress harmonics at specific frequencies and cannot achieve the ideal harmonic suppression effect in the case of rich harmonics. Additionally, PF may resonate with the impedance of the power system. And the parameters of PF may also change with the surrounding working conditions [6]. Active power filter has high compensation accuracy and excellent real-time performance, and it can dynamically compensate for various harmonics of the power grid. Therefore, it has become the most important means of harmonic suppression in the power grid [7,8]. The introduction of three-level technology has further expanded the range of applications for APF [9].

Active power filter is usually controlled by pulse width modulation (PWM). Therefore, the output current often contains a large amount of high-frequency switching harmonics [10]. Using a larger L filter is easy to absorb these high-frequency harmonics, but this method often increases the required cost, system size and controller bandwidth [11,12]. The LCL filter obtains better high-frequency harmonic attenuation ability by adding a small amount of additional capacitor filter without changing the size of the inductor filter [13]. Therefore, the LCL filter gradually replaces the L filter.

The working principle of the active power filter with an LCL filter is generating a compensation current with equal amplitude and opposite phase according to the detected command current, and then the compensation current is injected into the power grid for compensation. This requires that APF can track harmonic current in real-time, and the harmonic current tracking performance of APF mainly depends on the control strategy of current inner loop [14,15]. At present, the control method often used in active power filter is to convert the AC signal into a DC signal in the synchronous rotating coordinate system through the Clark transform, and then the proportional integral (PI) controller is used for tracking control. However, this control method has static tracking error and current coupling between the $d$-axis and $q$-axis, which ultimately leads to low harmonic current tracking accuracy [16]. Therefore, some scholars have developed a composite control strategy by combining repetitive control with PI control. This method takes into account the advantages of repetitive control and PI control, so it has high steady-state accuracy and fast dynamic performance. However, APF system still has the problem of incomplete decoupling [17]. The proportional resonant (PR) control based on the inner membrane principle has a high gain at its resonance point, so it can achieve zero steady-state error control. An adaptive proportional resonant controller is proposed. The fuzzy logic algorithm is used

as the adaptive gain mechanism of the proportional resonant controller. Even in the case of nonlinear load mutation, the APF system can still maintain high harmonic suppression performance [18]. In [19], the dominant pole elimination (DPE) algorithm is applied to the PR controller, which enhances the dynamic performance of the PR controller. In [20], an improved phase compensation method is proposed, and it can provide sufficient phase margin for the PR controller to keep the system stable. Quasi-proportional resonant (QPR) control is a special case of PR control. Compared with PR control, the bandwidth of QPR control at resonance point is significantly increased, so the stability of the system is improved [21]. In [22], a quasi-proportional resonant control with leading phases is proposed. The algorithm integrates harmonic detection and control into the current closed-loop, which not only simplifies the harmonic detection process, but also improves the steady-state accuracy and dynamic response speed. Based on the study of quasi-proportional resonant control, a multi-quasi proportional resonant (MQPR) control is proposed. The multi-quasi proportional resonant control can automatically change the resonant gain according to the inductive load and capacitive load. Therefore, it improves the compensation performance of APF [23]. In [24], a compound control combining fast repetitive control and PR control is proposed. This method can improve the compensation accuracy while maintaining the dynamic performance of proportional resonant, and it effectively reduces the tracking error of the system. In [25], an advanced repetitive control (ARC) of shunt active power filter based on Taylor series expansion is proposed. This method still maintains high harmonic suppression performance when the grid frequency and load change. In [26], a fractional-order repetitive control algorithm based on Lagrange interpolation is proposed, which can effectively improve the dynamic performance of APF without affecting the steady-state accuracy. In [27], a hybrid control method based on proportional integral control and fast repetitive control (FRC) is proposed. This method has advantages in steady-state compensation accuracy, load change adaptability and dynamic performance. In [28], an implementation scheme of virtual inertial APF based on PR and repetitive control is proposed. The composite control improves the harmonic suppression ability of APF, and the frequency droop can adjust the capacitor voltage amplitude when the system frequency fluctuates. Ref. [29] improves the QPR controller and proposes an adaptive quasi-proportional phase compensation resonant control strategy. This control strategy not only enhances the system's robustness to grid frequency variations, but also enhances the system's ability to suppress specific harmonics. In summary, the use of QPR control or related composite control strategies can effectively improve the tracking performance of the system, but there is a current coupling problem in the above control methods, and if the decoupling control cannot be achieved, the $d$-axis and $q$-axis will interfere with each other and affect the control performance [30].

Currently, the most common decoupling control strategy is feedforward decoupling control,which can achieve complete decoupling of the $d$-axis and $q$-axis. But the use of feedforward decoupling control will increase the cost of sensors [31]. In [32], a decoupling control strategy based on generalized inverse system method is proposed. This strategy can configure the system poles by adjusting the generalized inverse parameters and have a good decoupling control effect. Howerver, it is difficult to design the controller in practical application. Active disturbance rejection control (ADRC) is a control method that does not rely on precise system models, and it has strong anti-disturbance capabilities in the face of changes in system model parameters or unknown disturbances [33,34]. In [35], a double closed-loop control strategy based on ADRC and fractional order proportional integral differential (PID) is proposed, which has stronger robustness and higher compensation accuracy. Due to the fact that there are many ADRC parameters and no direct relationship between them, a linear

active disturbance rejection control is usually used to replace it in practical application. Linear active disturbance rejection control (LADRC) inherits the advantages of ADRC and reduces the number of parameters that need to be tuned by linearizing them [36]. In [37], LADRC control is applied to the voltage outer loop control of single-phase APF. This method still has good anti-interference performance and dynamic response speed under nonlinear load mutation. An improved linear active disturbance rejection control is proposed. By introducing a new error to adjust the total disturbance and successfully applied to the voltage outer loop, the anti-interference ability of the system is improved [38]. Ref. [39] proposed a repetitive and LADRC control, where LADRC compensates for the current coupling part as an internal disturbance and decouples it to improve the harmonic current tracking accuracy of the APF system. By connecting deadbeat control with repetitive and LADRC in series, deadbeat repetitive LADRC control is formed. This control strategy can suppress odd harmonics, shorten the cycle of repetitive control and further improve the steady-state current tracking accuracy of APF system [40]. In [41], PI control is used to replace the linear state error feedback (LSEF) control. Then, a PI and linear extended state observer (LESO) decoupling control method is formed. Due to the use of fourth-order LESO, the control method can achieve high-performance decoupling control of the system, but there are many parameters to be tuned.

Combining the advantages of the above control strategies, this paper proposes a dual closed-loop control strategy suitable for neutral-point-clamped (NPC) type active power filter. The inner current loop adopts repetitive quasi-proportional resonant with linear active disturbance rejection control, where the former repetitive QPR control is used to eliminate steady-state error and improve harmonic current tracking accuracy, and the latter LADRC control is used to observe and compensate current coupling to achieve high-performance decoupling control. Compared with the PI control with feedforward decoupling control, the repetitive quasi-proportional resonant with linear active disturbance rejection control not only enhances the steady-state and dynamic performance of APF, but also reduces the sensor cost. The voltage outer loop adopts QPR control to improve the dynamic performance of the system under load changes and maintain the stability of the DC-side voltage. Finally, the effectiveness of the proposed double closed-loop control strategy is verified by real-time digital simulation system (RTDS).

The second part of this paper establishes and analyzes the mathematical model of APF. In the third part, the current inner loop is analyzed and designed. In the fourth part, the voltage outer loop is analyzed and designed. In the fifth part, the feasibility of the proposed control strategy is verified by RTDS online simulation. Finally, the conclusion is given.

## The establishment and analysis of APF mathematical model

### The establishment of APF mathematical model

Fig 1 is the topology of neutral-point-clamped three-level active power filter with an LCL filter. In the diagram, $u_{ga}$, $u_{gb}$, and $u_{gc}$ represent the three-phase grid voltage, and $i_{fa}$, $i_{fb}$, and $i_{fc}$ represent the harmonic current. $L_{l1}$, $L_{l2}$, and $C_{c1}$ are the inverter side filter inductance, the grid side filter inductance and the filter capacitance of the LCL filter, respectively. $R_{c1}$ is the damping resistance. $i_{l1a}$, $i_{l1b}$, and $i_{l1c}$ are the output current of the inverter, and $i_{l2a}$, $i_{l2b}$, and $i_{l2c}$ are the output current of the inverter to the grid. $C_1$ and $C_2$ are the DC-side voltage dividing capacitor, $i_n$ is the current flowing into the voltage dividing capacitor, and $V_{dc}$ is the voltage of DC-side.

In order to facilitate the subsequent analysis, the parasitic resistance and damping resistance $R_{c1}$ of the LCL filter are ignored. According to Kirchhoff's first and second laws and

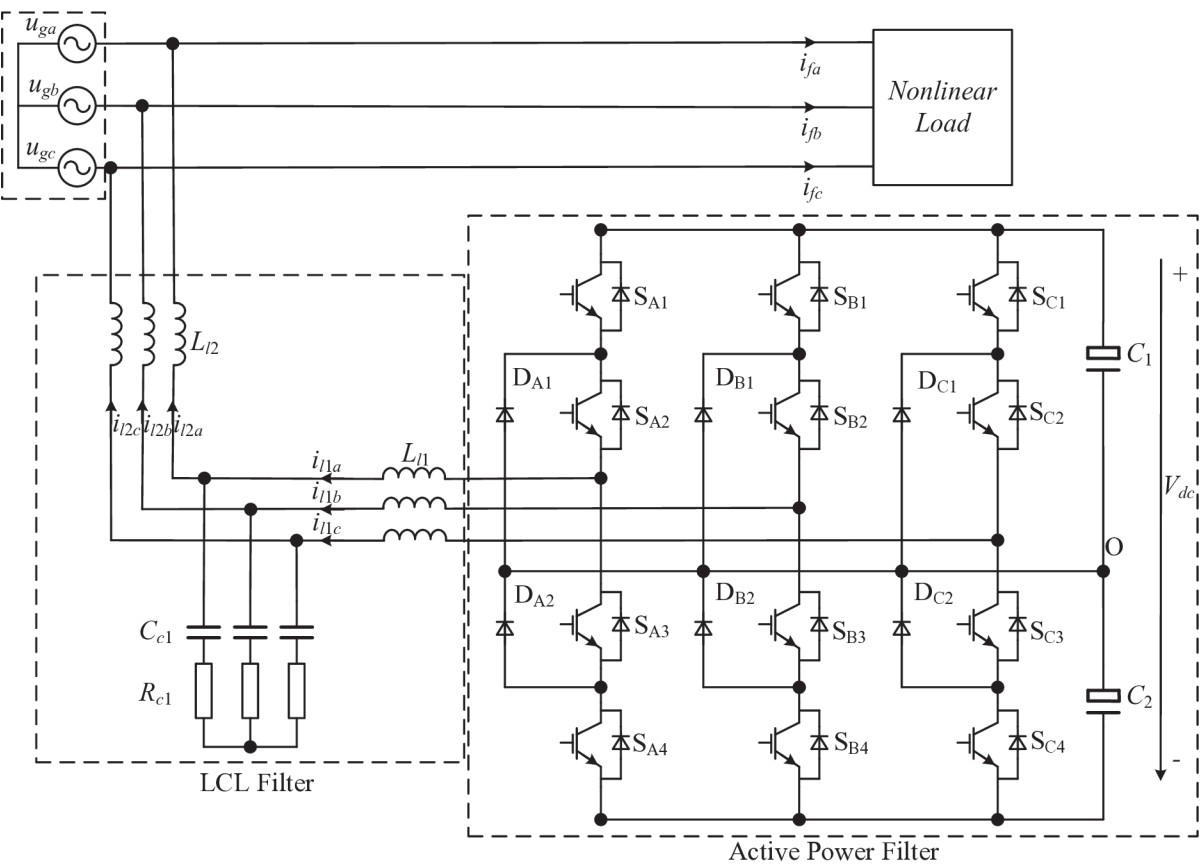

**Fig 1. Topology of the three-level active power filter with an LCL filetr.**

combining Fig 1, the mathematical model of LCL-type APF in *abc* coordinate system can be obtained:

$$
\begin{cases}
L_{l1}\dfrac{di_{l1a}}{dt} + u_{c1a} - u_{l1a} = 0 \\[4pt]
L_{l1}\dfrac{di_{l1b}}{dt} + u_{c1b} - u_{l1b} = 0 \\[4pt]
L_{l1}\dfrac{di_{l1c}}{dt} + u_{c1c} - u_{l1c} = 0 \\[4pt]
C_{c1}\dfrac{du_{c1a}}{dt} - i_{l1a} + i_{l2a} = 0 \\[4pt]
C_{c1}\dfrac{du_{c1b}}{dt} - i_{l1b} + i_{l2b} = 0 \\[4pt]
C_{c1}\dfrac{du_{c1c}}{dt} - i_{l1c} + i_{l2c} = 0 \\[4pt]
L_{l2}\dfrac{di_{l2a}}{dt} + u_{ga} - u_{c1a} = 0 \\[4pt]
L_{l2}\dfrac{di_{l2b}}{dt} + u_{gb} - u_{c1b} = 0 \\[4pt]
L_{l2}\dfrac{di_{l2c}}{dt} + u_{gc} - u_{c1c} = 0
\end{cases}
\tag{1}
$$

In Eq (1), $u_{l1a}$, $u_{l1b}$, and $u_{l1c}$ represent the output voltage of the inverter. $u_{c1a}$, $u_{c1b}$, and $u_{c1c}$ represent the voltage of the filter capacitance.

From Eq (1), it can be seen intuitively that in the three-phase stationary *abc* coordinate system, the mathematical model of the system can intuitively show the relationship between the various quantities, and the physical meaning is clear. However, in this model, the inverter side current, the grid side current and the filter capacitor voltage are all sinusoidal AC quantities. It is difficult to control the sinusoidal AC quantity directly. And, it is impossible to achieve no static error control. In order to facilitate the controller design, *d–q* coordinate system is often used for analysis and design in engineering. Therefore, the mathematical model of LCL-type APF in *d–q* coordinate system can be obtained by Park transformation of Eq (1) [42]:

$$\begin{cases} L_{l1}\dfrac{di_{l1d}}{dt} = \omega L_{l1} i_{l1q} - u_{c1d} + u_{l1d} \\[6pt] L_{l1}\dfrac{di_{l1q}}{dt} = -\omega L_{l1} i_{l1d} - u_{c1q} + u_{l1q} \\[6pt] C_{c1}\dfrac{du_{c1d}}{dt} = \omega C_{c1} u_{c1q} + i_{l1d} - i_{l2d} \\[6pt] C_{c1}\dfrac{du_{c1q}}{dt} = -\omega C_{c1} u_{c1d} + i_{l1q} - i_{l2q} \\[6pt] L_{l2}\dfrac{di_{l2d}}{dt} = \omega L_{l2} i_{l2q} - u_{gd} + u_{c1d} \\[6pt] L_{l2}\dfrac{di_{l2q}}{dt} = -\omega L_{l2} i_{l2d} - u_{gq} + u_{c1q} \end{cases} \tag{2}$$

From Eq (2), the LCL-type APF system is a complex system with high order, strong coupling and many variables. Designing a controller for such a system is quite complicated. Therefore, the principle of LCL filter will be analyzed below, and the system will be simplified to facilitate controller design.

## Analysis of the LCL filter

Because the LCL filter has equal three-phase parameters and the three-phase power grid is symmetrical, the circuit in Fig 1 can be simplified. Taking phase a as an example, the simplified APF single-phase equivalent circuit diagram is shown in Fig 2.

In this structure, the LCL filter exhibits resonance phenomena, where the amplitude-frequency curve shows a resonance peak, and the phase-frequency curve shows a negative step of $-180°$, which will cause the system to be unstable. Therefore, this paper adds passive damping to suppress resonance in the system.

Combining Fig 2 and Eq (1), the relationship between the output voltage $u_{l1}$ of the inverter and the current $i_{l2}$ flowing from the inverter to the grid side can be obtained [43,44]:

$$G_{LCL}(s) = \frac{i_{l2}}{u_{l1}} = \frac{R_{c1}s + 1}{L_{l1}L_{l2}C_{c1}s^3 + (L_{l1} + L_{l2})R_{c1}s^2 + (L_{l1} + L_{l2})s} \tag{3}$$

From Eq (3), the active power filter with grid side current feedback control is a third-order system. Controlling a high-order system with a low-order controller is often difficult

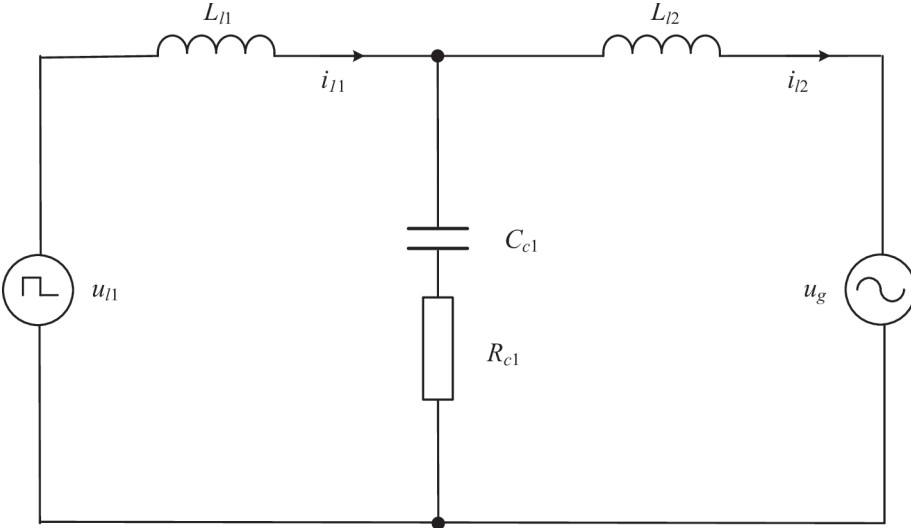

**Fig 2. The single-phase equivalent circuit diagram of APF.**

to achieve optimal control effects, and designing a high-order controller depends on the precise model of the controlled object [45,46]. For this reason, the Padé approximation method is often used to approximate the first-order system model to the third-order system model, which helps to achieve a balance between control effect and design complexity. By using the Padé approximation method, Eq (3) can be reduced to Eq (4) [47,48].

$$G_{LCL}(s) = \frac{i_{l2}}{u_{l1}} \approx \frac{1}{(L_{l1} + L_{l2})s} = \frac{1}{L_l s} = G_L(s) \tag{4}$$

From Eq (4), it can be obtained that after the Padé approximation method is used to reduce the order of the system model, the relative order of the system model becomes first-order [49]. At this time, the LCL filter can be approximately equivalent to an L filter. Therefore, the mathematical model of APF can be simplified as Eq (5) [50,51].

$$\begin{cases} L_l \dfrac{di_{l2d}}{dt} = u_{l1d} - u_{gd} + wL_l i_{l2q} \\ L_l \dfrac{di_{l2q}}{dt} = u_{l1q} - u_{gq} - wL_l i_{l2d} \end{cases} \tag{5}$$

As shown in Eq (5), there is voltage and current coupling in the simplified APF mathematical model. In order to achieve the ideal control effect, decoupling control should be considered in the subsequent design of the controller.

Since the reliable operation of APF needs to meet the two conditions that APF system can accurately track harmonic current and DC-side voltage stability, this paper adopts double closed-loop control system, in which the current inner loop adopts repetitive QPR-LADRC control, and the voltage outer loop adopts QPR control. The current inner loop and the voltage outer loop will be designed separately below.

## Design and analysis of current inner loop

### Design and analysis of QPR control

 Due to the fact that traditional proportional resonant control cannot effectively respond to changes in grid frequency, so the QPR control is often used as an alternative in practical applications [52]. As a special case of PR control, QPR control adds a new zero point on the basis of PR control, so that the frequency characteristic curve near the resonant frequency changes from steep to smooth. This not only produces a higher amplitude gain near the resonant frequency, but also enhances the anti-interference of the system [53].

The transfer function of QPR control is [54]:

$$G_{QPR}(s) = k_{p1} + \frac{2k_{r1}w_{c1}s}{s^2 + 2w_{c1}s + w_{01}{}^2} \tag{6}$$

where, $k_{p1}$ is the proportional gain coefficient. $k_{r1}$ is the resonance factor. $w_{c1}$ is the sideband angular velocity of the controller, and $w_{01}$ is the resonance frequency.

Substituting the resonant frequency into the $G_{QPR}(s)$, the gain at the resonant frequency can be obtained as:

$$A_{QPR}(jw_{01}) = k_{p1} + k_{r1} \tag{7}$$

From Eq (7), it can be seen that the gain of the QPR controller at the resonance frequency $w_{01}$ is only related to $k_{p1}$ and $k_{r1}$. Therefore, the resonance gain can be improved by adjusting the values of $k_{p1}$ and $k_{r1}$, and then the static error can be eliminated to achieve no static error control. Keeping $w_{c1}$ constant, the Bode plot of the QPR controller with $k_{p1}$ and $k_{r1}$ changed is shown in Fig 3.

From Fig 3, it can be seen that the controller gain can be increased by increasing $k_{p1}$ and $k_{r1}$. Among them, increasing $k_{p1}$ increases the gain of the frequency band, while increasing $k_{r1}$ only increases the gain of the resonant frequency and its nearby frequency. Considering that too high gain may lead to system instability [55,56], this paper finally chooses $k_{p1} = 20$, $k_{r1} = 100$.

Below, the impact of $w_{c1}$ on the QPR controller is analyzed. Since the harmonic frequency of the power grid mainly faced in this experiment is 5th, 7th, and 11th harmonic. 5th harmonic is taken as an example for analysis, and $w_{01} = 500$ rad/s is selected. The Bode diagram of the QPR controller with $w_{c1}$ changed is shown in Fig 4.

From Fig 4, it can be seen that the sideband angular velocity $w_{c1}$ of the controller only affects the bandwidth and has no effect on the gain at the resonance. Furthermore, the larger $w_{c1}$ is set, the larger the bandwidth at the resonance frequency will be.

When specific subharmonics need to be compensated, different $w_{01}$ values need to be set for current tracking control, so multiple QPR controllers are generally required for control [57]. In this experiment, the nonlinear load generates harmonics mainly concentrated on the 5th, 7th, 11th, and more than 20th harmonic content is less. Therefore, the QPR controller designed in this paper mainly enhances the ability of APF to suppress the 5th, 7th, and 11th harmonics. The control block diagram of the QPR controller is shown in Fig 5.

Where, $w_{01} = 2\pi f$. $f$ is the fundamental frequency. $i_{l2}{}^*$ represents the command current signal, and $e_{l2} = i_{l2}{}^* - i_{l2}$ represents the tracking error.

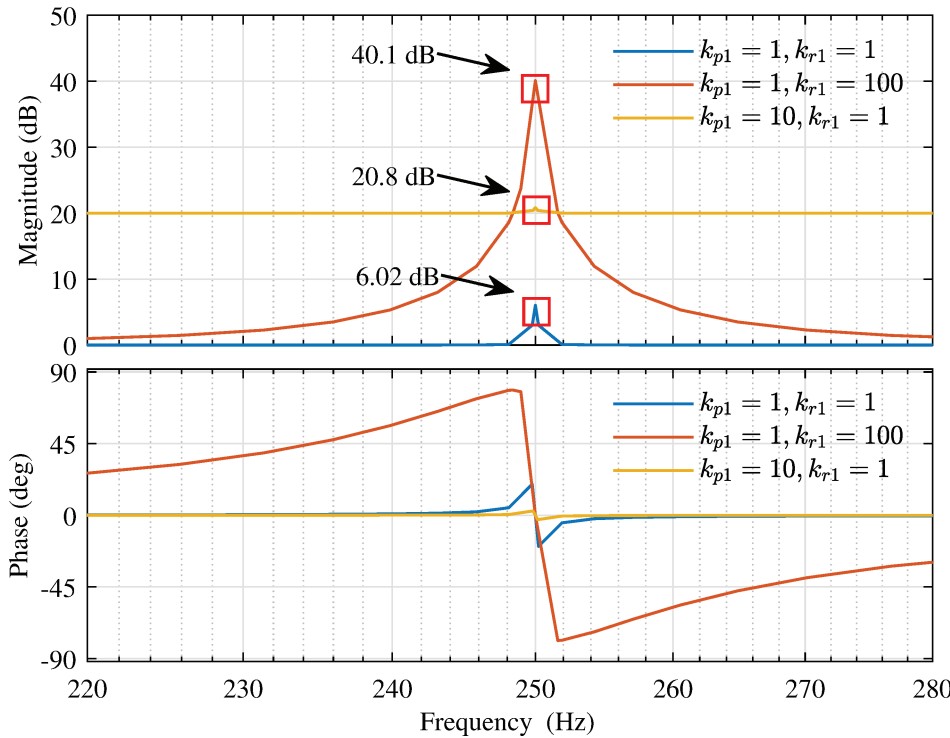

**Fig 3. Bode diagram of the QPR controller.**

## Design and analysis of repetitive QPR control

The inner membrane theory is the core of the repetitive controller, and the essence of this theory is to implant the mathematical model of the dynamic characteristics of the command signal and the disturbance signal into the control loop to form a high-precision feedback control system [58]. Therefore, the use of repetitive control can greatly reduce the steady-state tracking error of AC system and improve the harmonic suppression capability of APF. The block diagram of the repetitive control structure based on the inner membrane in the $z$ domain is shown in Fig 6.

In Fig 6, the repetitive controller can be divided into two parts: the inner membrane structure and the compensator. $Q(z)$ and $z^{-N}$ constitute the inner membrane structure. $K_r z^k S(z)$ is the compensator, and the controlled object is $G_{l2}(s)$. $Q(z)$ represents a low-pass filter used to attenuate the integration effect, and a constant slightly less than 1 is usually used for substitution to simplify the inner membrane [59]. By increasing $Q(z)$, the inner membrane gain can be increased, thereby improving the steady-state accuracy of the system, but it will cause a decrease in system stability. In order to ensure the stability of the system, $Q(z) = 0.95$ is selected in this paper. $N = T/T_s = 400$ represents the number of sampling points in a basic period [60]. Here, $T_s$ represents the sampling time in the $z$ domain, and $T$ represents the signal period to be generated. $e_r(z) = i_{l2}{}^*(z) - i_{l2}(z)$ represents the tracking error in $z$ domain.

The design of the compensator is the most important part. $K_r$ is the gain coefficient of repetitive control, and the value range is usually $0 < K_r < 2$ [61]. In this paper, $K_r = 1$ is selected. $z^k$ is the leading link, which is used to compensate the phase delay of the controlled

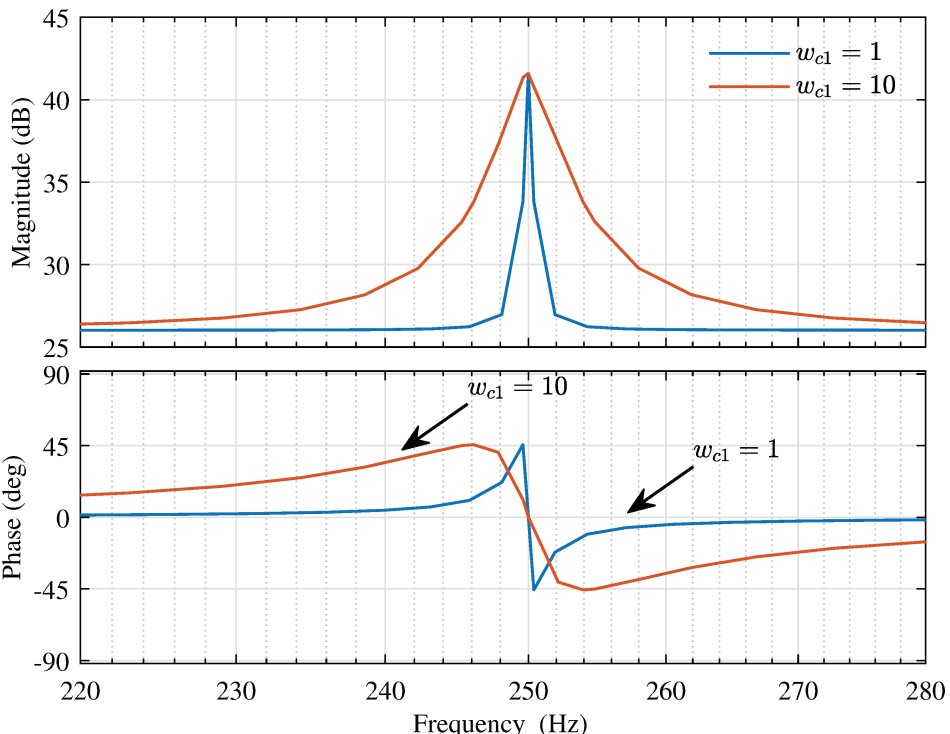

**Fig 4. The Bode diagram when $w_{c1}$ changes.**

object $G_{l2}(z)$ in the low frequency band [62]. In this paper, $k = 4$ is selected. $S(z)$ is a second-order low-pass filter, and its main function is to suppress the high-frequency interference of the system and improve the stability of the system. Therefore, selecting an excessively high or low filter cutoff frequency is meaningless. An excessively high cutoff frequency may cause system instability, while a low cutoff frequency will reduce the APF system's ability to suppress high-frequency harmonics [63,64].

The expression of the second-order low-pass filter in the $s$ domain is shown in Eq (8). Considering that the harmonics of the nonlinear load in this experiment are mainly below 20 times, this paper selects the cutoff frequency $w_n$ as $2\pi \times 2500$ rad/s and the damping ratio $\xi$ as 0.707.

$$S(s) = \frac{w_n^2}{s^2 + 2\xi w_n s + w_n^2} = \frac{(2\pi \times 2500)^2}{s^2 + 2 \times 0.707 \times 2\pi \times 2500 s + (2\pi \times 2500)^2} \tag{8}$$

After discretization by using the zero-order holder method, Eq (8) becomes Eq (9).

$$S(z) = \frac{0.2098z + 0.1443}{z^2 - 0.9753z + 0.3294} \tag{9}$$

Fig 7 shows the Bode diagram of the LCL filter $G_{LCL}(s)$ with feedback grid side current.

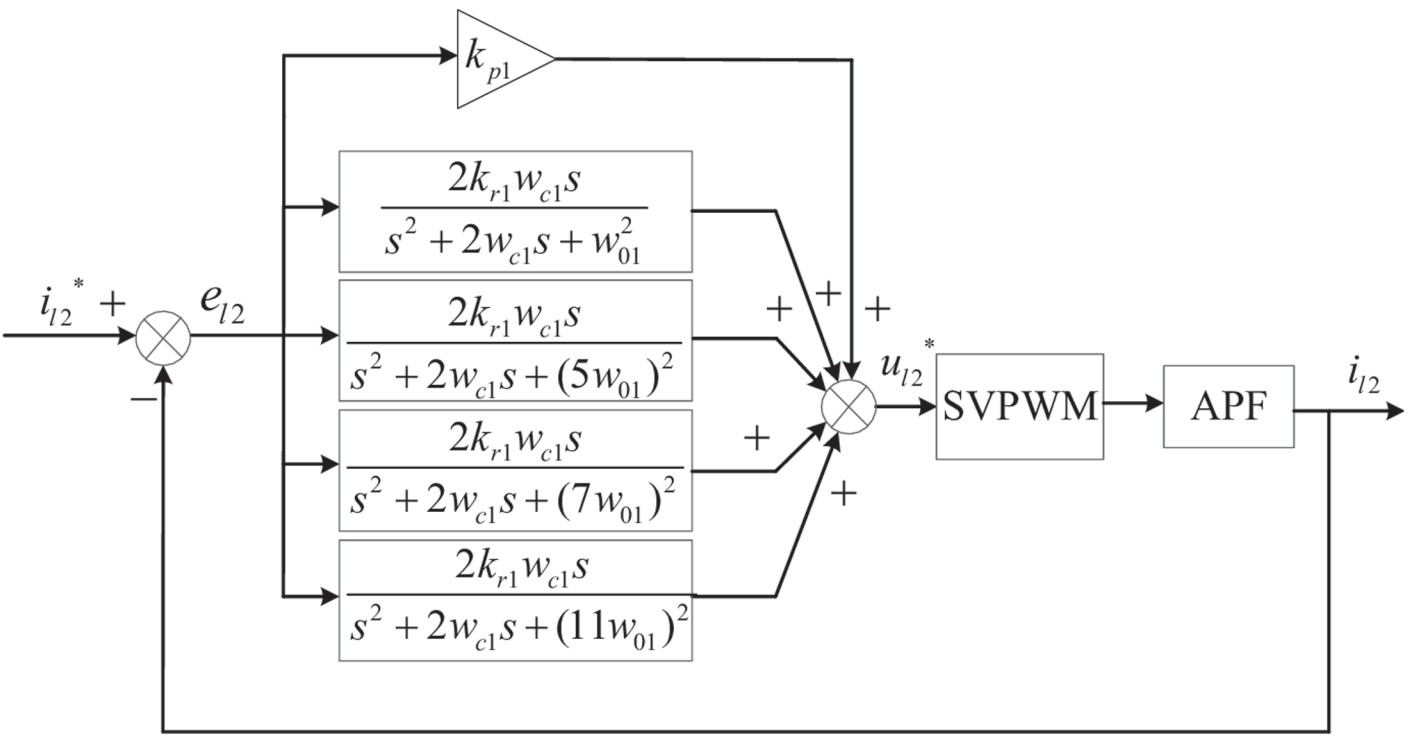

**Fig 5. Schematic diagram of the QPR control.**

From Fig 7, its frequency characteristics are similar to single inductor in the low frequency band (below 500). Therefore, this paper uses the differential link with inertia to offset, and the system transfer function after offset is shown in Eq (10).

$$G_{l2}(s) = G_{LCL}(s) \times \frac{Ks}{\tau s + 1}$$

$$= \frac{2.31 \times 10^{-8} s^2 + 0.0021s}{4.4 \times 10^{-16} s^4 + 6.82 \times 10^{-12} s^3 + 4.431 \times 10^{-7} s^2 + 0.0021s}$$

(10)

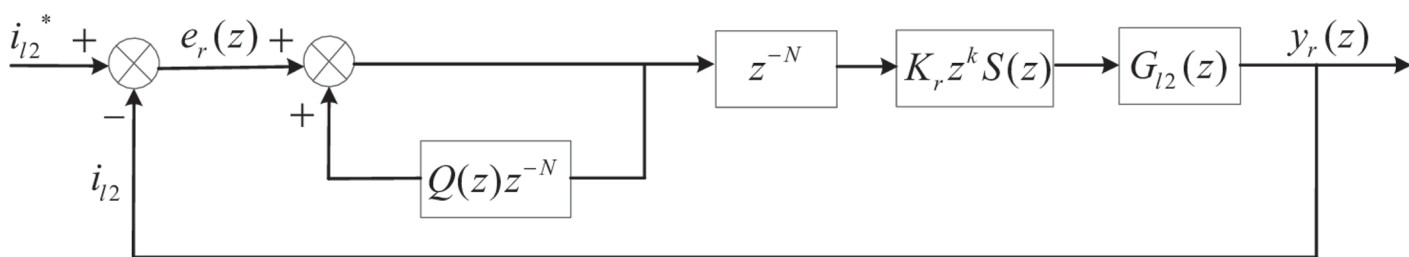

**Fig 6. The structure diagram of the repetitive control.**

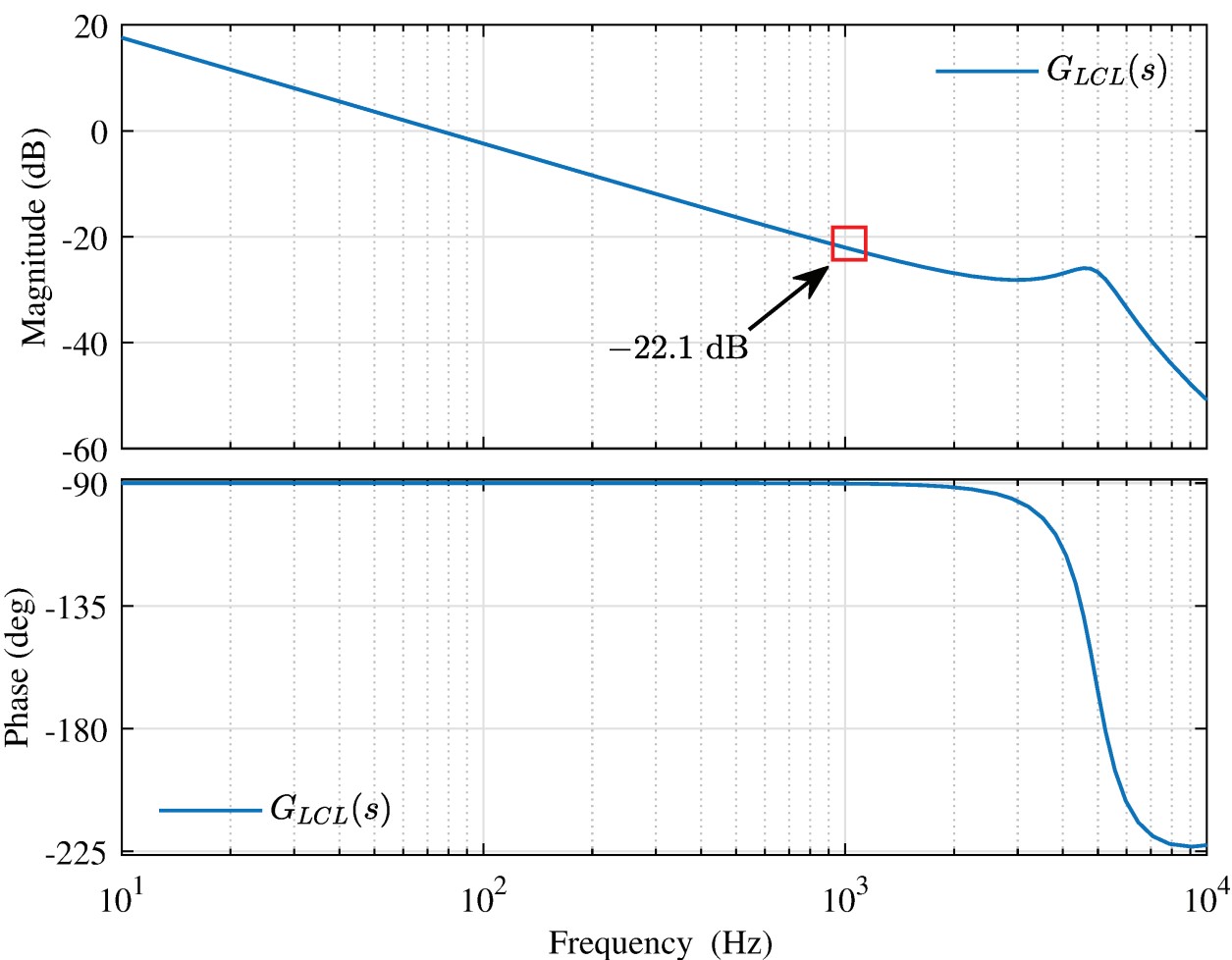

**Fig 7. Bode diagram of the LCL filter.**

According to Fig 6, the characteristic equation of the system with repetitive control can be obtained.

$$1 - Q(z)z^{-N} + G_{l2}(z)S(z)z^4 z^{-N} = 0 \tag{11}$$

According to the small gain theorem [65], it can be concluded that the system is stable when the characteristic roots of the system characteristic equation fall in the unit circle on the z plane (i.e., $|zi| < 1$). Then, Eq (11) is simplified to Eq (12).

$$\left| Q(z) - G_{l2}(z)S(z)z^4 \right| < 1 \tag{12}$$

Finally, the stability of the repetitive controller is further verified by the Nyquist curve. The Nyquist curve of $G_{l2}(z)S(z)z^4$ is shown in Fig 8.

Since $Q(z)$ is set to 0.95 in this article, a unit circle with a radius of 1 is drawn around the point (0.95, 0). From Fig 8, it can be seen that the Nyquist curve of the $G_{l2}(z)S(z)z^4$ repetitive

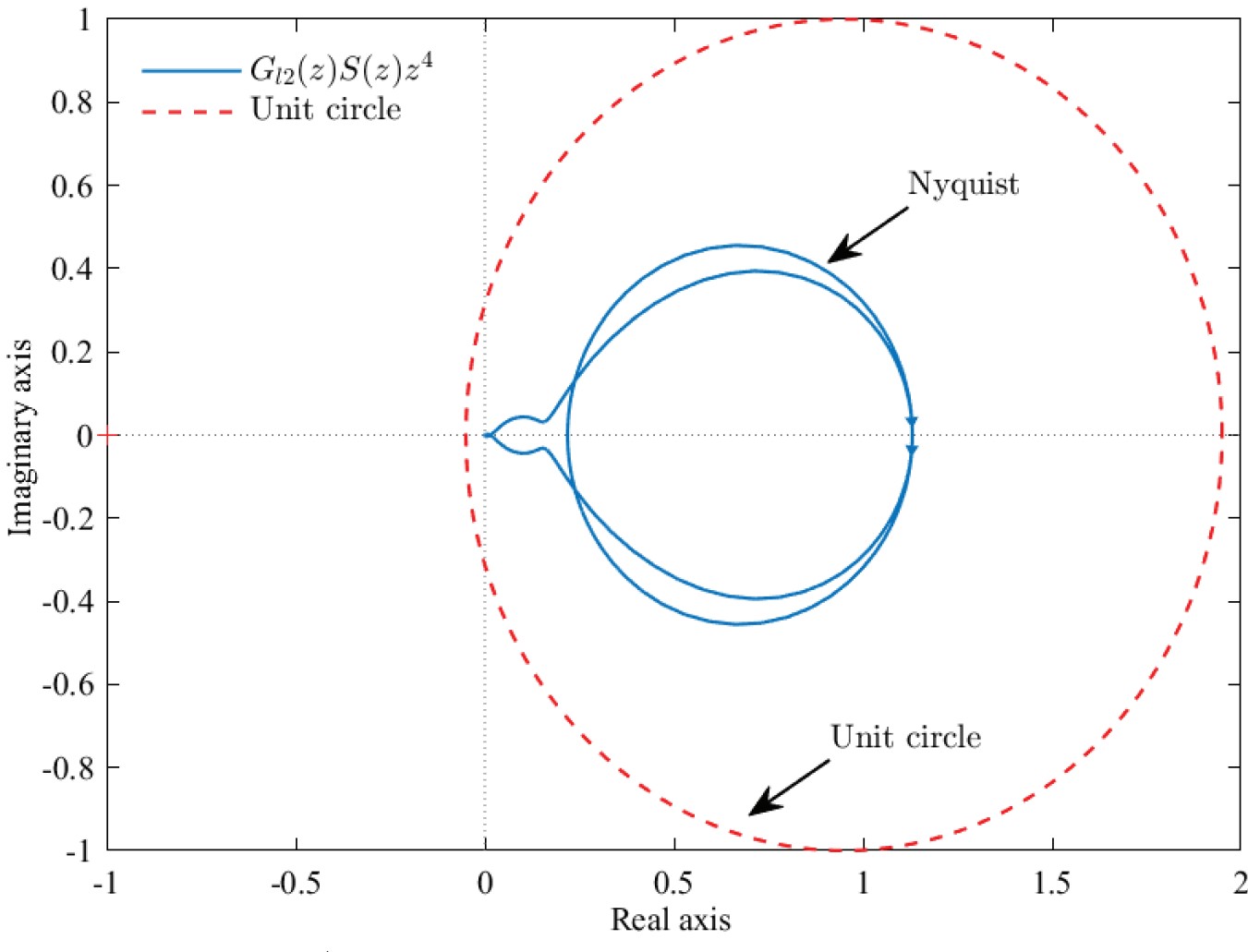

**Fig 8. Nyquist curve of the $G_{l2}(z)S(z)z^4$.**

control system is contained within the unit circle, therefore,it satisfies the stability condition and deduces that the repetitive control system is stable.

By combining QPR control with repetitive control in parallel, a repetitive QPR composite control can be obtained. The composite control takes into account the advantages of QPR and repetitive control, which can quickly and stably track the AC component in the system without static error. Thereby, it can improve the harmonic suppression effect of APF system. The control block diagram of the repetitive QPR compound control is shown in Fig 9.

## Design and analysis of LADRC control

From Eq (4), it can be concluded that after the Padé approximation method is applied, the transfer function of the APF system with LCL filter is reduced to first-order, so a first-order LADRC controller can be used for control.

The first-order LADRC controller mainly consists of the tracking differentiator (TD), the linear state error feedback rate (LSEF), and the linear extended state observer. Among them, the main function of TD module is to smoothly transition the given input by extracting the

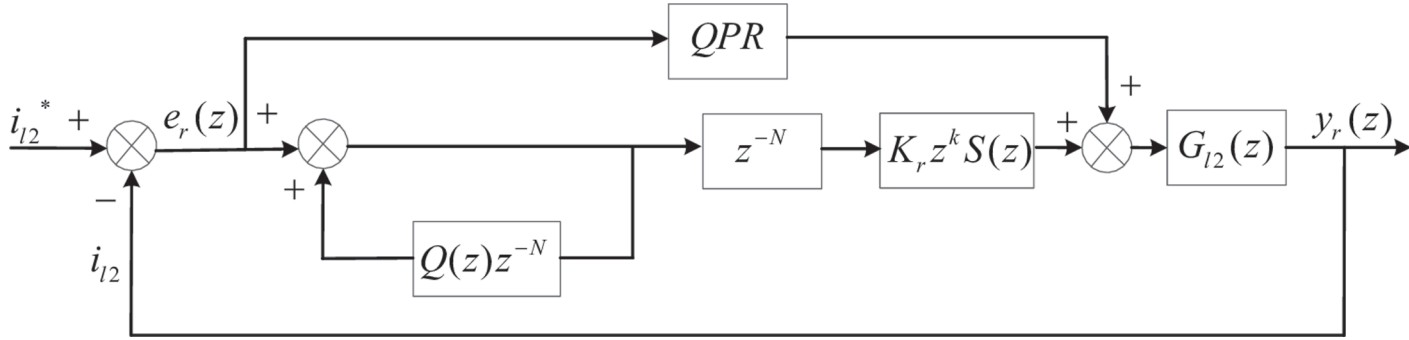

**Fig 9. The structure diagram of repetitive QPR control.**

dynamic characteristics of the input signal. LESO is responsible for observing the total disturbance of the system, and LSEF is responsible for performing differential control of the input signal and output signal error. In order to improve the response speed of the system, the TD module is not used in this paper [66]. The control block diagram of the first-order LADRC without the TD module is shown in Fig 10 [67].

In Fig 10, $u^*$ is the reference signal that the controller needs to track. $i_{l2}$ is the current output by APF to the grid side. $w_q$ is the undetermined external disturbance, APF is the controlled object of the controller, and $u_q$ is the voltage output by the controller. $e_q = u^* - i_{l2}$ represents the tracking error.

According to the LADRC control block diagram shown in Fig 10, Eq (5) can be rewritten in the following form:

$$\begin{cases} \dfrac{di_{l2d}}{dt} = b_q u_{l1d} + f_{qd}(i_{l2d}, i_{l2q}, u_{gd}, w_q) \\ \dfrac{di_{l2q}}{dt} = b_q u_{l1q} + f_{qq}(i_{l2d}, i_{l2q}, u_{gq}, w_q) \end{cases} \tag{13}$$

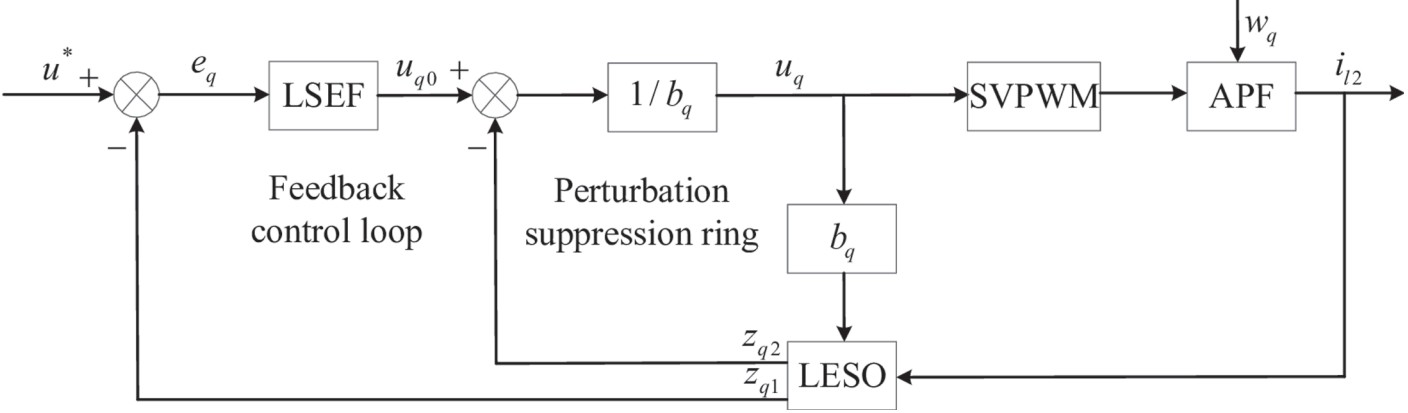

**Fig 10. The structure diagram of first-order LADRC control.**

Specifically, $b_q = 1/L_l$. $f_{qd}$ and $f_{qq}$ represent the sum of internal and external disturbances in the APF model, and it includes the current coupling disturbance between $i_{l2d}$ and $i_{l2q}$. The model error disturbance after the order reduction using the Padé approximation method and the external uncertain disturbance $w_q$. Considering that the $d$–$q$ axis has the same LADRC control structure, for the sake of convenience, only the $d$-axis will be analyzed in the following.

Two state variables $x_{q1}$ and $x_{q2}$ are defined, where $x_{q1}$ represents the current $i_{l2}$ output by APF to the grid side. $x_{q2}$ represents the total internal and external disturbances $f_q$ of APF system. Suppose that $f_q$ is differentiable, Eq (13) can be expressed as a state space equation.

$$\begin{cases} \dot{x}_q = Ax_q + Bu_i + E\dot{f}_q \\ y_q = Cx_q \end{cases} \tag{14}$$

where,

$$x_q = \begin{bmatrix} x_{q1} \\ x_{q2} \end{bmatrix}, A = \begin{bmatrix} 0 & 1 \\ 0 & 0 \end{bmatrix}, B = \begin{bmatrix} b_q \\ 0 \end{bmatrix}, C = \begin{bmatrix} 1 & 0 \end{bmatrix}, E = \begin{bmatrix} 0 \\ 1 \end{bmatrix}, u_i = u_{l1}, y_q = i_{l2}$$

**Design and analysis of LESO.** From Eq (14), the second-order linear extended state observer can be designed.

$$\dot{z}_q = (A_q - LC_q)z_q + B_q u_i + LCx_q \tag{15}$$

where

$$z_q = \begin{bmatrix} z_{q1} \\ z_{q2} \end{bmatrix}, A_q = A, B_q = B, C_q = C, L = \begin{bmatrix} \beta_1 \\ \beta_2 \end{bmatrix}$$

In Eq (15), $z_{q1}$ is the sensing value of the current output from APF to the grid, which is used to track $i_{l2}$. While $z_{q2}$ is the observation value of the total disturbance of APF system, which is used to track $f_q$. $\beta_1$ and $\beta_2$ represent the observer gain of LESO.

From Eq (15), the convergence rate of the linear extended state observer is controlled by the poles of $A_q - LC_q$. Therefore, according to the bandwidth parameter tuning method [68], as long as all the poles of $A_q - LC_q$ are configured at $-w_0$, the observation error of the system will converge to zero, and the fast non-overshoot tracking of the reference signal can be realized. The corresponding expected characteristic polynomial is shown in Eq (16).

$$\lambda(s) = \left| sI - (A_q - LC_q) \right| = (s + w_0)^2 \tag{16}$$

According to Eq (16), the elements in the output error feedback matrix $L$ can be obtained as:

$$\begin{cases} \beta_1 = 2w_0 \\ \beta_2 = w_0^2 \end{cases} \tag{17}$$

where, $w_0$ is the LESO bandwidth.

**Design and analysis of LSEF.** LSEF can feed back the state observations of the system observed by LESO to the controller. Therefore, the total output of the controller can be

expressed as:

$$u_q = \frac{u_{q0} - f_q}{b_q} \qquad (18)$$

Among them, $u_{q0}$ is the control input without disturbance compensation. When there is no observation error between the observed value $z_{q2}$ and the actual value $f_q$, the system can be further simplified.

$$\dot{y}_q = u_{q0} + f_q - z_{q2} \approx u_{q0} \qquad (19)$$

It can be seen from Eq (19) that the system has been corrected to an integral series structure, so only proportional controller can be designed to eliminate the static error.

$$u_{q0} = k_{pp}(v_q - z_{q1}) \qquad (20)$$

Specifically, $v_q = u^*$ represents the signal to be tracked. $k_{pp}$ is the gain of the controller, and the controller bandwidth $w_c = k_{pp}$ is often used.

By substituting Eq (20) into Eq (19), the closed-loop transfer function from the input reference signal to the final output can be obtained.

$$G_{vl}(s) = \frac{w_c}{s + w_c} \qquad (21)$$

Where, $w_c$ is the controller bandwidth, and $w_0$ is the observer bandwidth. It can be seen from Eq (21) that the system will become a first-order low-pass filter after full compensation for internal and external disturbances.

## Design and analysis of repetitive QPR-LADRC control

The control block diagram of the repetitive QPR-LADRC controller is shown in Fig 11. At the front-end, repetitive control and QPR control are parallelly connected to form a repetitive QPR control. The repetitive QPR control can not only eliminate the periodic error contained in the stable closed-loop to improve the steady-state tracking accuracy of the APF system, but also enhance the APF's ability to suppress specific harmonics. At the back-end, LADRC is connected in series to form repetitive QPR-LADRC composite control. LADRC controller can observe and compensate the current coupling part to achieve high-performance decoupling control. Thus, it can further improve the steady-state and dynamic performance of APF.

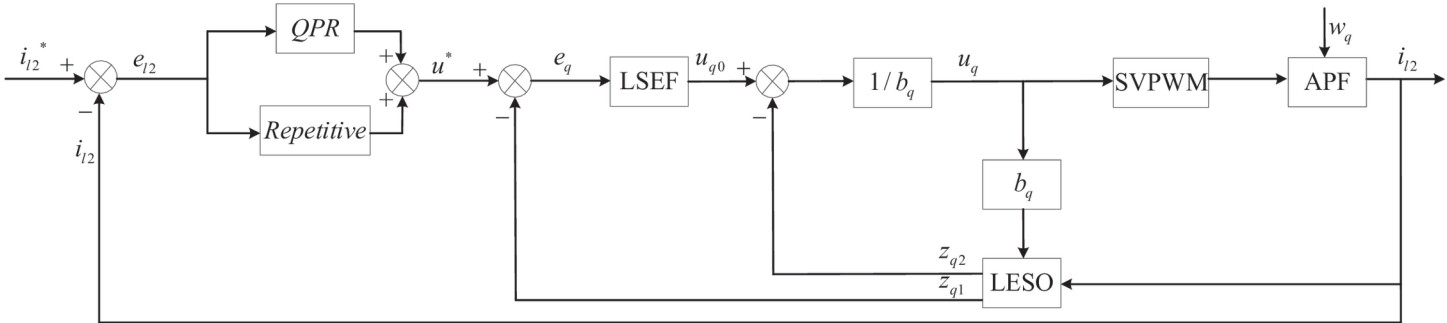

**Fig 11. The structure diagram of repetitive QPR-LADRC control.**

As shown in Fig 11, the error $e_{l2}$ is calculated by comparing the reference given value $i_{l2}{}^*$ and the actual output value $i_{l2}$, and then the system control signal with disturbance is generated by QPR and repetitive controller processing. Finally, the disturbance compensation is carried out by LADRC to form the final system control signal.

## Parameter tuning

A first-order LADRC controller requires three tunable parameters: the proportional coefficient $b_q$, the controller bandwidth $w_c$, and the observer bandwidth $w_0$. The observer bandwidth affects the accuracy of state estimation, the higher the bandwidth, the higher the estimation accuracy, but a too large bandwidth will amplify high-frequency noise. Therefore, when choosing the observer bandwidth, the balance between state estimation accuracy and high-frequency noise immunity needs to be considered [70]. The controller bandwidth is usually selected according to the stability time required by the system [71].

The specific parameter tuning process is as follows:

1. In general, the observer bandwidth $w_0$ and the controller bandwidth $w_c$ should be maintained in the relationship of $w_0 = w_c \sim 10w_c$.
2. For a system with known model parameters, $b_q$ should be set according to the actual model parameters. In this case, $b_q = 1/L_l$. If the model parameters are unknown or more complex, a larger $b_q$ can be chosen initially.
3. Let $w_0 = 10w_c$, and $w_0$ is gradually increased until the system oscillates, then $w_0$ is backtracked to the stable state.
4. After confirming the value of $w_0$, $w_c$ is gradually increased to ensure the stability of the system and to obtain better dynamic performance and anti-interference performance as much as possible.
5. Gradually reduce $b_q$, cycle the previous steps, record the dynamic performance indicators under different parameters, and select the optimal parameters.

## Design and analysis of voltage outer loop

Whether the DC-side voltage is stable or not affects the harmonic compensation effect of APF. The PI controller has a simple structure and is easy to control, so it is the most commonly used DC-side voltage control strategy. However, PI control has limited dynamic modulation ability in the face of sudden changes in harmonic load [69]. The quasi-proportional resonant control can enhance the dynamic modulation ability of the system and improve the control effect of the voltage outer loop. Therefore, this paper uses QPR controller to control the voltage outer loop. The specific control block diagram is shown in Fig 12.

In Fig 12, the DC-side voltage error $e_{dc}$ is derived by comparing the DC-side voltage reference value $V_{dc}{}^*$ with the actual DC-side voltage $V_{dc}$, and then the active power current $i_{dc}{}^*$ required for compensation is generated by processing the error signal through the QPR controller. The active power current $i_{dc}{}^*$ is then superimposed onto the $d$-axis as the control signal for the current inner loop, and finally the APF is controlled through the current inner loop to achieve stable DC-side voltage.

From Fig 12, the DC-side voltage is regulated by the active command current generated by the QPR controller of the voltage outer loop:

$$i_{dc}{}^* = \left(k_{p2} + \frac{2k_{r2}w_{c2}s}{s^2 + 2w_{c2}s + w_{02}{}^2}\right)\left(V_{dc}{}^* - V_{dc}\right) \tag{22}$$

In Eq (22), $k_{p2}$ is positively correlated with the proportional gain of the system. If the value of $k_{p2}$ is too large, it will cause the system to oscillate. If the value is too small, it will reduce the anti-interference ability of the system. Therefore, this paper selects $k_{p2} = 4$. In order to reduce the harmonic content and steady-state error, $k_{r2} = 100$, $w_{02} = 100\pi$, and the control loop bandwidth angle $w_{c2} = 20$ are selected.

## Results and discussion

In order to verify the effectiveness of the proposed double closed-loop control strategy, RTDS is used for hardware-in-the-loop (HIL) simulation test and the simulation results are given. Compared with off-line simulation, HIL simulation can narrow the gap between the simulation system and the real system. In the process of HIL simulation, the power system studied is represented by the simulation model on the real-time simulation platform. The main control board is located outside the model and connected to the simulation model through the analog-digital board and the digital-template board. Because the HIL simulation test can simulate the real situation of the power system, the designer can reduce the dependence on the test environment. One of the biggest advantages of HIL is that designers can evaluate actual models without physical hardware [3,10–12]. Using the HIL platform can also save the cost of hardware verification and reduce the time and effort required for designers to develop and modify various applications. In addition, compared with hardware experiments, redesigning HIL experiments is faster and easier, so real-time testing can often be completed faster than hardware testing [72]. At present, HIL has become a modern technology that is often used in the design and verification of power electronic systems.

The picture of the RTDS hardware-in-the-loop simulation platform is shown in Fig 13. Among them, RTDS NovaCor is a dedicated hardware simulation platform for RTDS real-time simulation system. NovaCor's processing chip is IBM's POWER8, the processor frequency is 3.5GHz and have 10-core processor. Each standard chassis can choose 1-10 cores for expansion and up to 60 chassis co-simulation. Therefore, it can be configured according to the size of the user's multi-power system or the complexity of the power electronic device. RTDS GTDI is used to access trigger pulses. Photoelectric conversion is used to convert electrical signals into optical fiber signals. The main control board can generate feedback state according to the PWM signal and control command.

RSCAD is a software owned by RTDS, which is specially designed to connect with the hardware of RTDS simulator. RSCAD is a full-featured software package that provides users with all the functions they need from preparation to running simulation and observing and analyzing simulation results, and it need not any third-party software. In order to highlight the superiority of the double closed-loop control strategy proposed in this paper, an APF

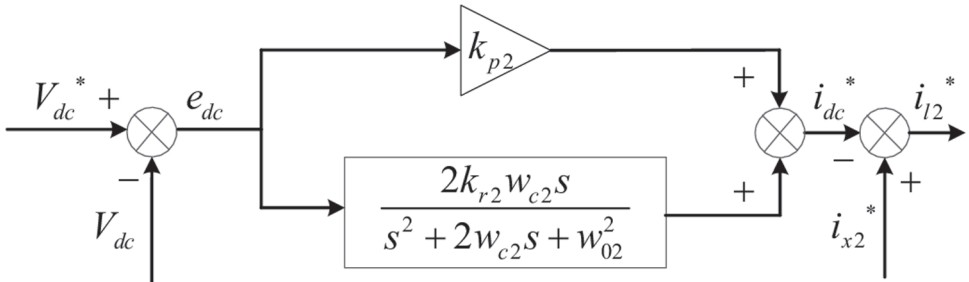

**Fig 12. Control strategy of the voltage outer loop.**

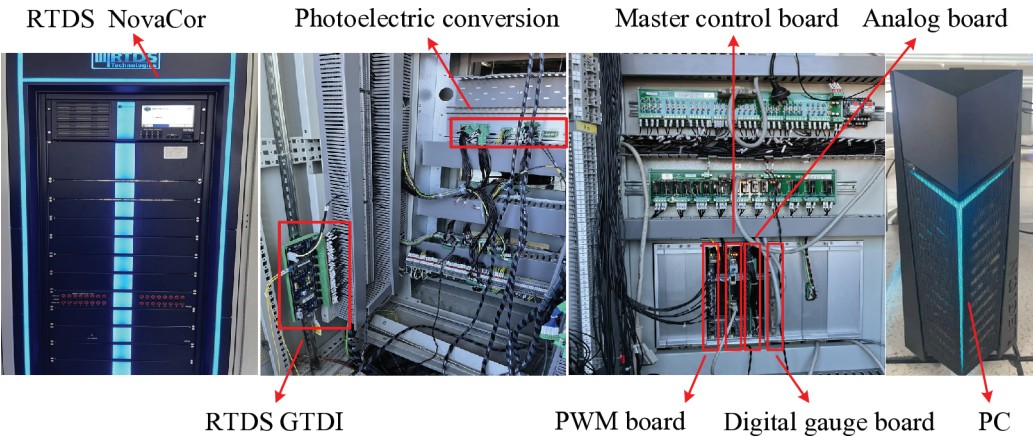

RTDS NovaCor    Photoelectric conversion    Master control board   Analog board

RTDS GTDI    PWM board    Digital gauge board    PC

**Fig 13. RTDS real-time simulation hardware platform.**

**Table 1. Simulation parameters of the APF system.**

| Parameters | Numerical Value | Parameters | Numerical Value |
|---|---|---|---|
| Grid voltage | 0.48 kV | Inverter side inductance | 2 mH |
| Grid side inductance | 0.1 mH | Filter capacitance | 11 uF |
| DC-side capacitance | 23.5 mF | DC-side bus voltage | 1 kV |
| Gride frequency | 50 Hz | Switching frequency | 20 kHz |
| Msximum compensation current | 0.15 kA | $b_q$ | 500 |
| $w_0$ | 7000 | $w_c$ | 700 |

model with double closed-loop PI control and LADRC inner PI outer loop with the same simulation parameters is built on RSCAD. Some of the simulation parameters are shown in Table 1.

Firstly, the steady-state performance of the control strategy is tested under a constant load. Figs 14A, 14D, 15A, and 15D use the repetitive QPR-LADRC control, Figs 14B, 14E, 15B, and 15E use PI control, and Figs 14C, 14F, 15C, and 15F use LADRC control.

From Figs 14A-14C, it can be seen that the three control strategies can track the harmonic current well in steady-state. From Fig 14D-14F, the deviation between the harmonic command current and the APF output current controlled by repetitive QPR-LADRC is the smallest, and the tracking accuracy is the highest. The tracking deviation of PI control is the largest, and the tracking accuracy is the lowest. And the tracking accuracy of LADRC control is slightly lower than that of repetitive QPR-LADRC control. In Figs 15A-15C, it can be seen more intuitively that after harmonic compensation, the phase a grid current waveform controlled by repetitive QPR-LADRC is smoother and has fewer burrs. In addition, it can be seen from Figs 15D-15F that the three control strategies can control the DC-side voltage well, and the fluctuation range is similar when the load is constant.

Secondly, the dynamic performance of the control strategy is analyzed. The APF is set to start compensation at 0.1 s, and the load changes abruptly at 0.4 s (Harmonic current suddenly becomes smaller). Figs 16A, 16D, 17A, 17D, 18A, 18D, 19A, and 19D use repetitive QPR-LADRC control. Figs 16B, 16E, 17B, 17E, 18B, 18E, 19B, and 19E use PI control. Figs 16C, 16F, 17C, 17F, 18C, 18F, 19C, and 19F use LADRC control.

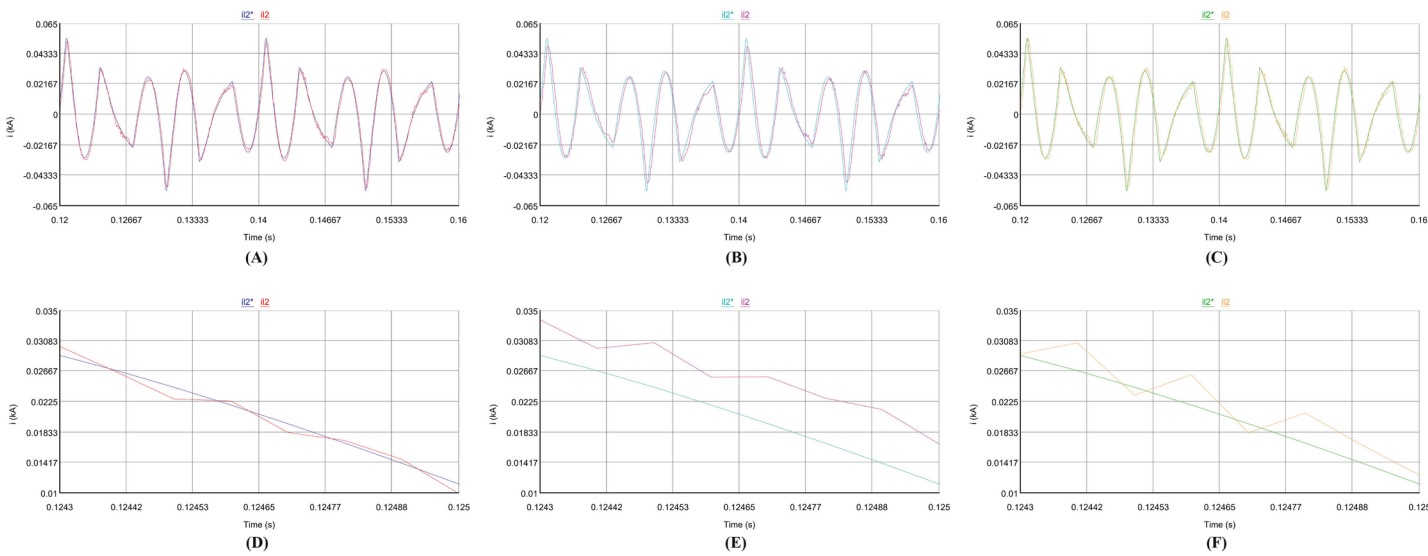

**Fig 14. Steady-state current tracking waveforms of the three control strategies.**

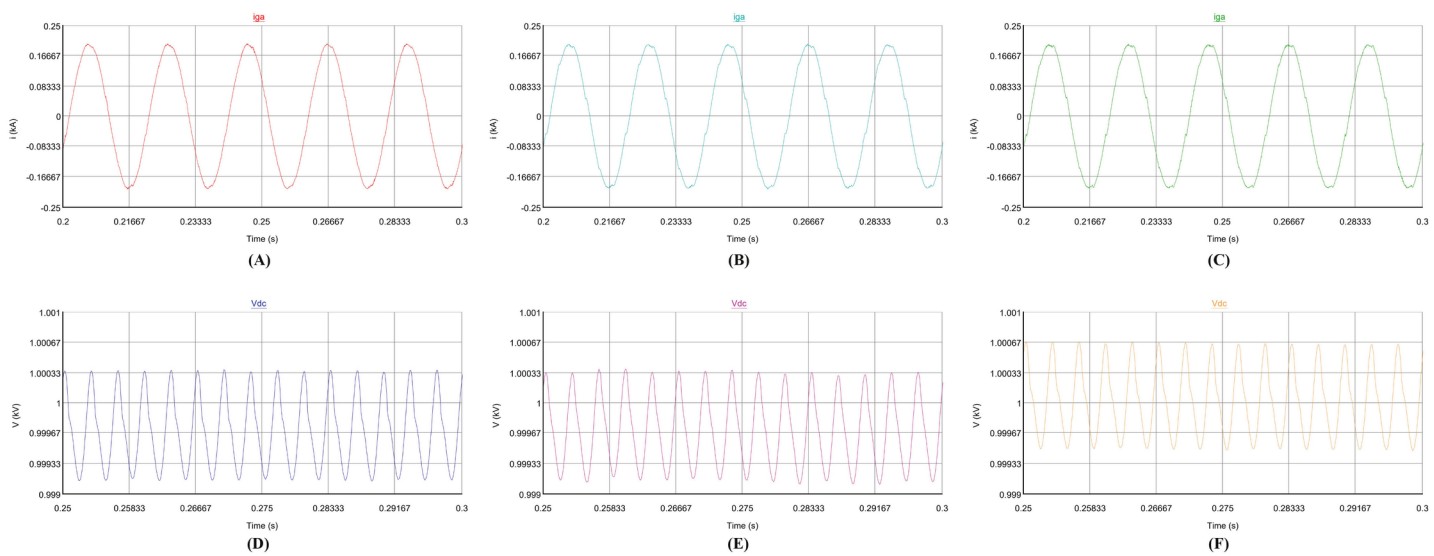

**Fig 15. Steady-state voltage and current waveforms of the three control strategies.**

From Figs 16A-16C, the traditional PI control needs 0.008 s to complete the harmonic current tracking, and the LADRC control needs 0.004 s to complete the harmonic current tracking, while the repetitive QPR-LADRC control can complete the tracking of the harmonic current at the start of the APF system. In Figs 16D-16F, it can also be seen that the traditional PI control has obvious grid current overshoot, and the grid current value is 0.2115 kA. The repetitive QPR-LADRC control and LADRC control can effectively suppress the grid current overshoot, and the grid current values are 0.1992 kA and 0.2048 kA, respectively. It can be seen that repetitive QPR-LADRC control has obvious advantages.

Then, the DC-side voltage is analyzed. From Figs 17A-17F, it can be seen that during the start-up process of APF, the APF system using traditional PI control has obvious voltage overshoot. The DC voltage value is 1.007 kV and it takes 0.13 s to reach stability. Although LADRC control can suppress voltage overshoot, it also has the problem of long stability time, which takes 0.129 s. The APF system using repetitive QPR-LADRC control can perform DC voltage control with almost no overshoot. The voltage value is 1.0002 kV and can quickly reach a stable state with a stable time of 0.05 s.

From Figs 18A-18C, when the load mutates at 0.4 s, it takes 0.018 s for the PI control to regain current tracking, while the LADRC control is similar to PI and takes 0.017 s. The repetitive QPR-LADRC control is significantly better than the other two control strategies, and it takes only 0.01 s. As shown in Figs 18D-18F, the grid current waveform of the repetitive QPR-LADRC control is smoother, and it can indicate that after the load mutation, the repetitive QPR-LADRC control can still maintain good harmonic suppression performance.

From Figs 19A-19C, all three control strategies have voltage overshoot when facing load perturbations. The APF system with repetitive QPR-LADRC and QPR dual closed-loop control has a voltage overshoot of 4 V, while the APF system with dual closed-loop PI control has a voltage overshoot of 5.6 V, and the APF system with LADRC and PI dual closed-loop control has a voltage overshoot of 6.1 V. This indicates that the repetitive QPR-LADRC and QPR dual closed-loop control strategy has an advantage. Furthermore, in Figs 19D-19F, the APF system with repetitive QPR-LADRC and QPR dual closed-loop control can recover stability faster, as it only takes 0.09 s, while the APF system with dual closed-loop PI control and LADRC and PI dual closed-loop control takes 0.113 s and 0.116 s, respectively.

In summary, the APF system with repetitive QPR-LADRC and QPR dual closed-loop control exhibits superior performance in both steady-state and dynamic performance.

Finally, the harmonic content of the grid current is analyzed, and the specific analysis results are shown in Table 2. From Table 2, it can be seen that when the grid current is 0.198 kA, the grid current total harmonic distortion (THD) is 17.76%, and there is serious waveform distortion. After filtering with the repetitive QPR-LADRC control APF system, the

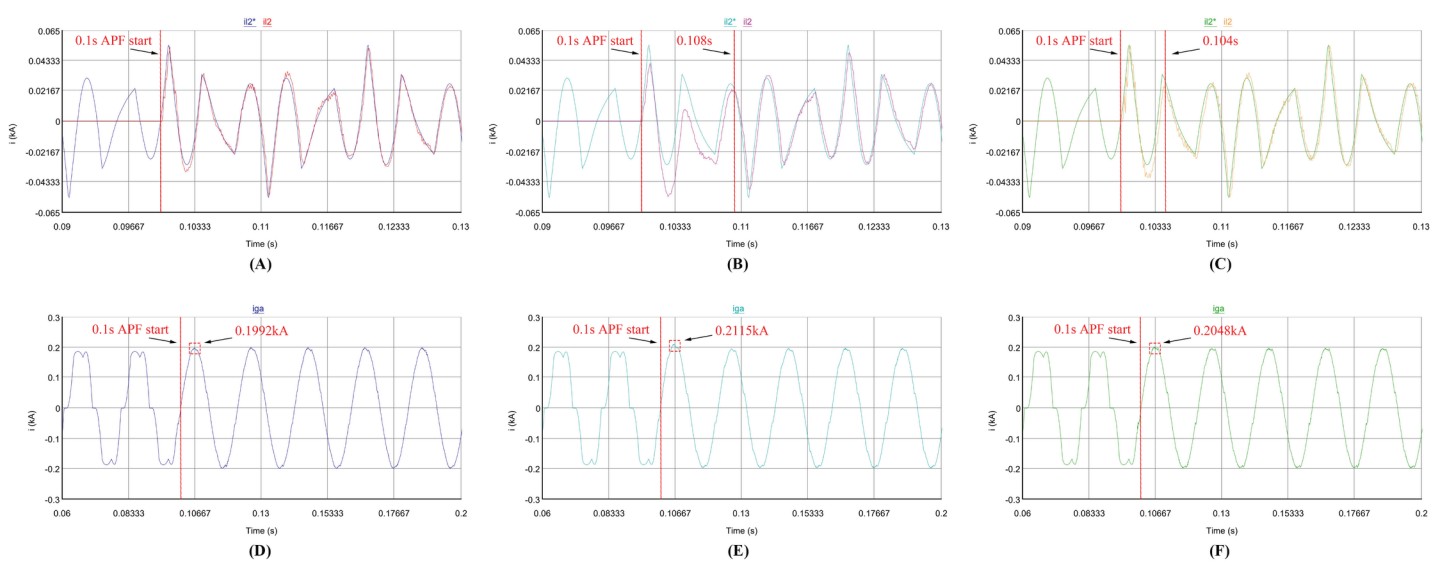

**Fig 16. The current waveforms of the three control strategies during APF start-up.**

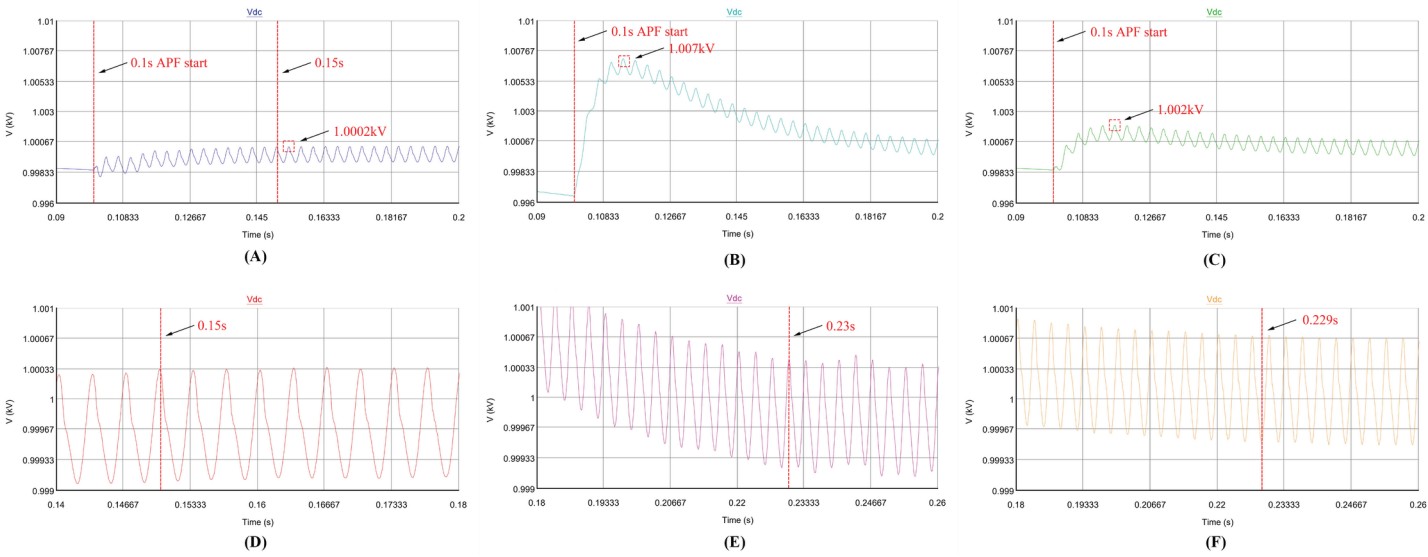

**Fig 17. The DC-side voltage waveforms of the three control strategies during APF start-up.**

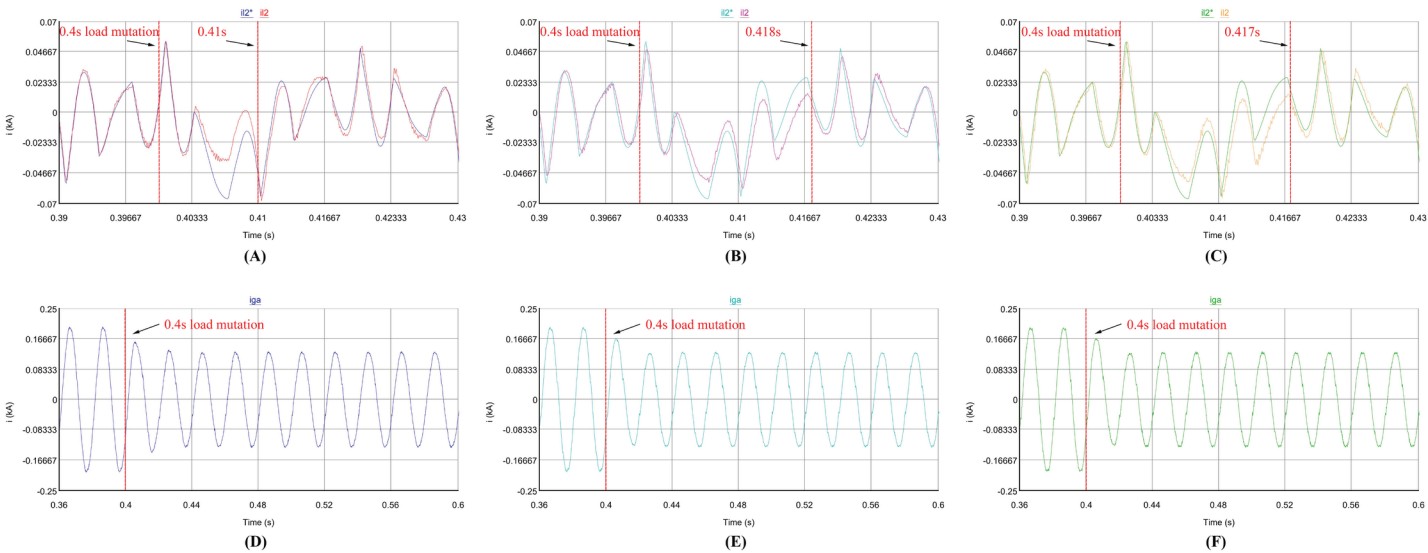

**Fig 18. The current waveforms of the three control strategies during load mutation.**

grid current THD is reduced to 1.64%, which is 1.84% lower than the traditional PI control APF system and 1.32% lower than the LADRC control APF system. When the load suddenly changes and the grid current drops to 0.13 kA, the grid current THD is 19.25%. After filtering with the repetitive QPR-LADRC of the APF system, the grid current THD is reduced to 2.32%, which is 2% lower than the traditional PI control APF system and 1.39% lower than the LADRC control APF system. In summary, the repetitive QPR-LADRC proposed in this paper can effectively improve the harmonic suppression capability of the APF.

Table 3 shows the compensation of the 5th, 7th and 11th harmonic currents in the grid before and after using three strategies. From Table 3, it can be seen that the 5th, 7th and 11th

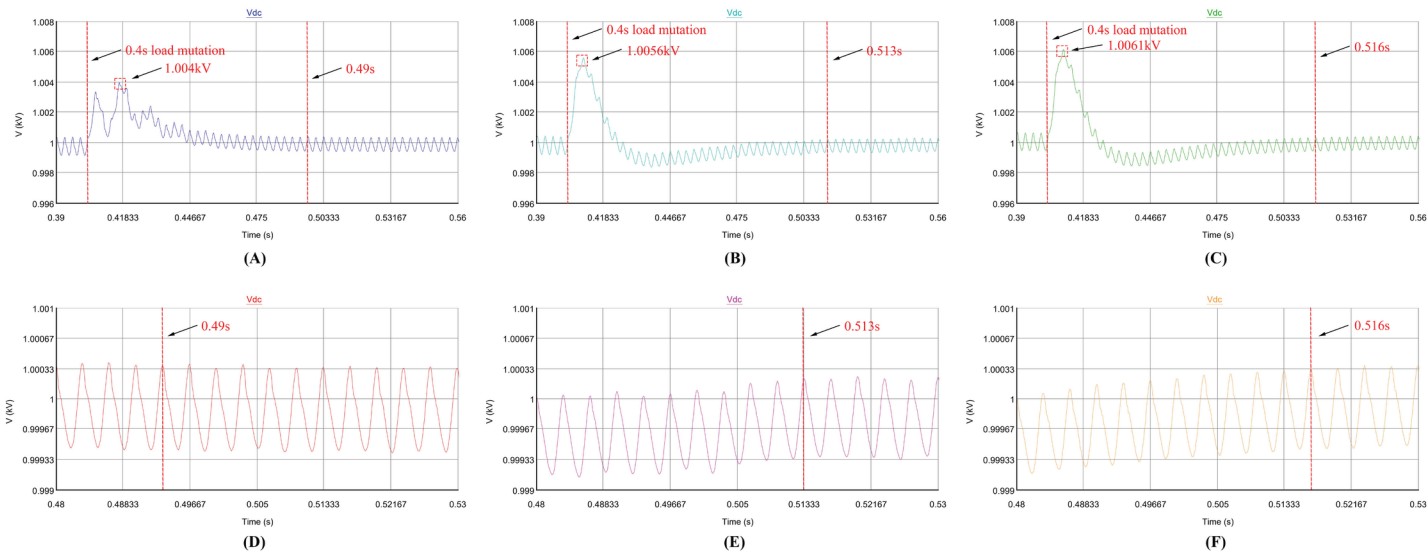

**Fig 19. The DC-side voltage waveforms of the three control strategies during load mutation.**

**Table 2. Comparison of harmonic current THD before and after filtering.**

| Load current | 0.198 kA | 0.13 kA |
|---|---|---|
| Pre-compensation | 17.76% | 19.25% |
| PI control | 3.48% | 4.32% |
| LADRC control | 2.96% | 3.71% |
| Repetitive QPR-LADRC | 1.64% | 2.32% |

**Table 3. Comparison of the 5th, 7th and 11th harmonic current THD before and after filtering.**

| Harmonic number | 5 | textbf7 | 11 | THD |
|---|---|---|---|---|
| Pre-compensation | 15.12% | 6.29% | 2.46% | 17.76% |
| PI control | 2.11% | 2.01% | 1.33% | 3.48% |
| LADRC control | 1.69% | 1.46% | 1.07% | 2.96% |
| Repetitive QPR-LADRC | 0.52% | 0.77% | 0.56% | 1.64% |

harmonic currents in the grid after using the repetitive QPR-LADRC is greatly reduced and significantly better than the traditional PI control and LADRC control.

## Conclusion

Since the traditional PI control can not track the harmonic current without static error and the APF system in the $d$–$q$ coordinate system has the problem of voltage and current coupling. In order to improve the tracking accuracy of APF to harmonic current and achieve high-performance decoupling, this paper proposes a double closed-loop control strategy based on repetitive QPR-LADRC and QPR control. The repetitive QPR-LADRC control is used for the current inner loop, where the repetitive QPR control is used to improve the tracking accuracy of APF, and the LADRC control is used for high-performance decoupling control. QPR control is used in the voltage outer loop to keep the DC-side voltage stable.

Finally, the proposed control strategy is validated through the RTDS online simulation platform, and the simulation results show that the dual closed-loop control strategy based on the repetitive QPR-LADRC and QPR control has the following advantages:

1. Compared with the traditional double closed-loop PI control and the double closed-loop control based on LADRC and PI control, the double closed-loop control strategy proposed in this paper has higher harmonic current steady-state tracking accuracy when the load is constant.
2. Compared with the traditional double closed-loop PI control and the double closed-loop control based on LADRC and PI control, the double closed-loop control strategy proposed in this paper has better dynamic tracking performance in the face of load mutation.
3. Compared with the traditional feedforward decoupling control, the repetitive QPR-LADRC control proposed in this paper can achieve high-performance decoupling while reducing sensor costs.
4. This method can coordinate the control of various frequency components and enhance the ability of APF to suppress specific harmonics.

The repetitive QPR-LADRC control proposed in this paper can solve the harmonic problem more effectively while saving the sensor cost, and it has good application value. However, compared with the traditional control method, the repetitive QPR-LADRC control needs more parameters to be determined, especially in the case of rich harmonic times. It is necessary to design a resonant controller for each harmonic separately, which leads to the complex structure of the controller and is not easy to be digitized. Therefore, the simplification of the controller still needs further research. In addition, in the process of LADRC parameter tuning, all poles are usually configured in the same position. Although this operation is convenient, it also weakens the performance of the controller. How to configure the poles is also the focus of subsequent research.

## Acknowledgments

The authors would like to thank the reviewers and the editor for their valuable comments and suggestions to improve this work.

## Author contributions

**Conceptualization:** Hongshi Wei, Xiaoguo Lv.

**Investigation:** Yifei Gao.

**Methodology:** Yifei Gao, Xiaoyang Wang.

**Software:** Hongshi Wei, Xiaoguo Lv.

**Supervision:** Liancheng Zhu.

**Validation:** Yifei Gao, Liancheng Zhu.

**Visualization:** Yifei Gao, Xiaoyang Wang.

**Writing – original draft:** Yifei Gao.

**Writing – review & editing:** Liancheng Zhu.

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
