## [Decision Letter · Decision Letter 0]

4 Nov 2024

PONE-D-24-40676Active power filter control strategy based on repetitive quasi-proportional resonant control with linear active disturbance rejection controlPLOS ONE

Dear Dr. Zhu,

Thank you for submitting your manuscript to PLOS ONE. After careful consideration, we feel that it has merit but does not fully meet PLOS ONE’s publication criteria as it currently stands. Therefore, we invite you to submit a revised version of the manuscript that addresses the points raised during the review process.

We look forward to receiving your revised manuscript.

Kind regards,

Akhtar Rasool, Ph.D.

Academic Editor

PLOS ONE

Journal Requirements:

3. Please note that PLOS ONE has spec6ific guidelines on code sharing for submissions in which author-generated code underpins the findings in the manuscript. In these cases, all author-generated code must be made available without restrictions upon publication of the work. Please review our guidelines at https://journals.plos.org/plosone/s/materials-and-software-sharing#loc-sharing-code and ensure that your code is shared in a way that follows best practice and facilitates reproducibility and reuse.

Additional Editor Comments:

The topology connections are not correct in figure 1.

Almost all figures require corrections too. The legends should be added properly. Even corrections are required in X and Y labeling. Since the figure numbers and captions are not provided with the figures, it is not easy to write/raise concerns in the specific figures. The results plots are also not provided with good resolution quality.

I dont think it is right to write modelling equations directly into a dq frame. It is therefore recommended strongly to also write the equations as per the topology provided before showing conversion to the dq frame, but it should be supported with at least needful discussion, if the authors dont feel it necessary to put all steps.

In line 148-149, controller gains are used, is there any specific reason/justification for selecting these?

Error equations could not be found, which are used for the formation/defining the controller.

The article has a lot of disconnection sections. So, I strongly advise the authors to revise (major) to complete/connect the missing parts to make the article more follow-able, understandable for the audience. The authors must understand that the article is also to be read by the new researchers for guidance and sequence of the matter presented, connections between the sections is very vital to be able to understand. However, currently it requires a lot of serious attention to fill the gaps besides some serious corrections.

Please address all concerns raised here and those raised by the reviewers point by point so it becomes convenient for the reviewers and the editor to see the particular sections to be able to help make a decision fast. Please note that the responses should be supported by the relevant changes in the article. Thanks

Looking forward to seeing your comprehensively revised article.

Reviewers' comments:

Reviewer's Responses to Questions

**Comments to the Author**

1. Is the manuscript technically sound, and do the data support the conclusions?

Reviewer #1: Yes

Reviewer #2: Yes

2. Has the statistical analysis been performed appropriately and rigorously? 

Reviewer #1: N/A

Reviewer #2: Yes

3. Have the authors made all data underlying the findings in their manuscript fully available?

Reviewer #1: Yes

Reviewer #2: No

4. Is the manuscript presented in an intelligible fashion and written in standard English?

Reviewer #1: Yes

Reviewer #2: Yes

5. Review Comments to the Author

Reviewer #1: This manuscript proposes an active power filter control strategy based on repetitive quasi-proportional resonant control with linear active disturbance rejection control. The idea is feasible and there is a scientific soundness in this paper. This is one of the good papers I have ever reviewed. However, there are some comments that need to be considered. The reviewer has the following concerns after careful reading:

1. There are a few writing issues. Also, the correct ‎punctuation rules are not observed in the text.

2. Some Nomenclature are not defined. Even the term is known, ‎any abbreviation should be defined for the first time, Then, no need to write the whole definition; please unify. The index terms are ‎not alphabetically ordered.‎ Could you ‎insert any reference for equations? It is more suitable to determine the ‎references ‎or explain the design procedure that was adopted to get these equations.

3. In the "Introduction" you claimed that “only 80% of harmonics can be filtered out at most by the passive power filters”, which is not accurate. So, the "Introduction" section should be made much more impressive and enriched with references by ‎adding and citing ‎the trends in the area of harmonics mitigation by passive power filters. E.g., High-frequency harmonics suppression in high-speed railway through magnetic integrated LLCL filter, Magnetic integrated double-trap filter utilizing the mutual inductance for reducing current harmonics in high-speed railway traction inverters, Magnetic integrated LLCL filter with resonant ‎frequency above Nyquist frequency, and Enhancing power quality of high-speed ‎railway traction converters by fully integrated T-LCL filter.‎ Some of These studies, which are published in the same journal PLOS ONE, are recent and discussing the latest techniques. Moreover, you claimed that “the higher the harmonic order, the less the harmonic current content, so it is usually only necessary to consider within 25 times” and only focused on 5th, 7th and 11th harmonic suppression. But, actually, the big issue in grid-connected converters is the switching harmonics, which you did not consider in the paper. You need to reconsider it and compare the ability of the proposed active power filter with the passive power filters. Furthermore, you are using an LCL filter, which is enough to suppress the harmonics is designed properly. So, what is the benefit of the proposed technique?

4. How did you select the parameters values?

5. Figures 17(b), 18(b), 17(d), and 18(d) were not explained and mentioned in the text.

6. The HIL advantages should be explained like those in new references High-frequency harmonics suppression in high-speed railway through magnetic integrated LLCL filter and Magnetic integrated double-trap filter utilizing the mutual inductance for reducing current harmonics in high-speed railway traction inverters. Otherwise, the real experiments are required.

7. The conclusion should be point wise.

8. In the technical writing, it is not true to write “we, I, our, etc.”. Moreover, you are not only one author, so, it is not true to write “I, my, etc.”. In addition, it is not preferred to thank your institutions. They, of course, help you achieve your research because it is their duties.

9. Many of the references are old. A state-of-the-art (SotA) review is preferred to show the real advantages of the proposed method against the recent literatures. All the references have no DOI. The format of references needs to be unified. Moreover, references need to be written in a standardized way. Many references are conferences papers.

From the aforementioned comments, I suggest the authors do more work in this field.‎ In the reviewers' opinion, the presented paper is sufficiently prepared ‎as a professional research paper. I tried to help you collect these comments as ‎‎highlighted in the attached file. Please consider my specific comments above.‎

Reviewer #2: The proposed method, though effective, appears complex, and the manuscript could benefit from a discussion on the feasibility and limitations of implementing this strategy in real-world scenarios.

While the paper compares the proposed method to traditional PI control, it lacks comparisons with more recent or alternative advanced harmonic suppression techniques.

The LADRC and QPR controllers involve multiple parameters. Providing guidelines or a systematic approach for parameter tuning would make the methodology more accessible.

6. PLOS authors have the option to publish the peer review history of their article (what does this mean?). If published, this will include your full peer review and any attached files.

Reviewer #1: No

Reviewer #2: No

---

## [Author Response · Author response to Decision Letter 1]

20 Nov 2024

Re: Manuscript PONE-D-24-40676: “Active power filter control strategy based on repetitive quasi-proportional resonant control with linear active disturbance rejection control”.

We thank reviewers and the editorial board for the expertise, constructive feedback and foremost time spent in providing such thorough reviews, which greatly improves the quality of the manuscript further. All the reviewers’ comments (in blue/italic) have been responded in full in the revised manuscript, and also detailed below. The red highlight indicates the revision of the opinion of the academic editor, the blue highlight indicates the revision of the opinion of the first reviewer, and the green highlight indicates the revision of the opinion of the second reviewer, while also taking into account the grammatical improvement.

Dear Akhtar Rasool,

We quite appreciate your favorite consideration and the reviewers’ insightful comments concerning our manuscript entitled “Active power filter control strategy based on repetitive quasi-proportional resonant control with linear active disturbance rejection control” (ID: PONE-D-24-40676). Those comments are very valuable and helpful for improving the quality and readability of our paper, as well as the important guiding significance to our future researches. We have studied the comments carefully and have revised the paper in detail according to the reviewers’ comments. We hope this revision can meet with approval. The revised part is marked in different colors in the paper. Red represents the revision of the academic editor 's comments, blue represents the revision of the first reviewer 's comments, and green represents the revision of the second reviewer 's comments. And the main revisions corresponding to the reviewers’ comments have been answered in detail below and in the paper.

Kind regards,  

Dr. Liancheng Zhu

Liaoning University of Technology

Journal Requirements:

3. Please note that PLOS ONE has spec6ific guidelines on code sharing for submissions in which author-generated code underpins the findings in the manuscript. In these cases, all author-generated code must be made available without restrictions upon publication of the work. Please review our guidelines at https://journals.plos.org/plosone/s/materials-and-software-sharing#loc-sharing-code and ensure that your code is shared in a way that follows best practice and facilitates reproducibility and reuse.

Response:

Authors have taken on boards Editors suggestion and have revised the paper according to the requirements of the journal. The specific financial disclosure can be seen in the file named Financial Disclosure.pdf.

Funding Information: This paper is funded by the 2024 Fundamental Research Project of the Educational Department of Liaoning Province. Funding number is (No. LJ212410154019).

Comment (1): The topology connections are not correct in figure 1.

Response: Thanks for your professional suggestions. Authors have revised the topology shown in Fig 1.

Comment (2): Almost all figures require corrections too. The legends should be added properly. Even corrections are required in X and Y labeling. Since the figure numbers and captions are not provided with the figures, it is not easy to write/raise concerns in the specific figures. The results plots are also not provided with good resolution quality.

Response: Thanks for your professional suggestions. We have added X and Y labels according to the format of the paper published on PLoS ONE, and we have updated the results and checked the images using PACE. At the same time, the figure numbers and captions of the figure were provided at the time of submission.

Comment (3): I dont think it is right to write modelling equations directly into a dq frame. It is therefore recommended strongly to also write the equations as per the topology provided before showing conversion to the dq frame, but it should be supported with at least needful discussion, if the authors dont feel it necessary to put all steps.

Response: Thank you for your suggestion. We think it is a very good suggestion. In the introduction of APF mathematical model, we added and analyzed the mathematical model in abc coordinate system and the unsimplified mathematical model in dq coordinate system.

Comment (4): In line 148-149, controller gains are used, is there any specific reason/justification for selecting these?

Response: Thank you for your question. The reasons for selecting these parameters are as follows. Increasing and can effectively improve the controller gain to achieve zero steady-state error tracking. Among them, is for all frequency bands, and is for the resonant frequency and its nearby frequencies. In order to facilitate readers ' understanding, we added Bode plots under different parameters and quote new references for readers ' in-depth understanding.

Comment (5): Error equations could not be found, which are used for the formation/defining the controller.

Response: Thanks for your sincere suggestions. The authors have added error equations.

Comment (6): The article has a lot of disconnection sections. So, I strongly advise the authors to revise (major) to complete/connect the missing parts to make the article more follow-able, understandable for the audience. The authors must understand that the article is also to be read by the new researchers for guidance and sequence of the matter presented, connections between the sections is very vital to be able to understand. However, currently it requires a lot of serious attention to fill the gaps besides some serious corrections.

Response: Thanks for your sincere suggestions. The author have revised the whole article and added the missing part ( highlighted in red blue and green font ).

Reviewer 1:

This manuscript proposes an active power filter control strategy based on repetitive quasi-proportional resonant control with linear active disturbance rejection control. The idea is feasible and there is a scientific soundness in this paper. This is one of the good papers I have ever reviewed. However, there are some comments that need to be considered. The reviewer has the following concerns after careful reading:

We thank the dear reviewer for the constructive suggestions. Reviewer’s valuable comments provide an important direction for us to revise our manuscript, which have been answered in detail and improved the manuscript greatly as follows:

Comment (1): There are a few writing issues. Also, the correct ‎punctuation rules are not observed in the text.

Response: The authors have worked hard to revise this manuscript in detail, we sincerely hope that it can achieve your request. Special thanks to you for your professional comments.

Comment (2): Some Nomenclature are not defined. Even the term is known, ‎any abbreviation should be defined for the first time, Then, no need to write the whole definition; please unify. The index terms are ‎not alphabetically ordered.‎ Could you ‎insert any reference for equations? It is more suitable to determine the ‎references ‎or explain the design procedure that was adopted to get these equations.

Response: Authors thank the reviewer’ sincere suggestions. We have defined terms that are first used in the text, reordered the index entries, and we have added references and analysis to explain the equations. Specific changes can be seen in the blue and red font part of the paper.

Comment (3): a. In the "Introduction" you claimed that “only 80% of harmonics can be filtered out at most by the passive power filters”, which is not accurate. So, the "Introduction" section should be made much more impressive and enriched with references by ‎adding and citing ‎the trends in the area of harmonics mitigation by passive power filters. E.g., High-frequency harmonics suppression in high-speed railway through magnetic integrated LLCL filter, Magnetic integrated double-trap filter utilizing the mutual inductance for reducing current harmonics in high-speed railway traction inverters, Magnetic integrated LLCL filter with resonant ‎frequency above Nyquist frequency, and Enhancing power quality of high-speed ‎railway traction converters by fully integrated T-LCL filter.‎ Some of These studies, which are published in the same journal PLOS ONE, are recent and discussing the latest techniques.

Response: Thanks for your sincere suggestions. The papers recommended by the reviewer are very good and valuable, so we have added them to the references. We have modified the introduction and made a more comprehensive discussion of passive filters. Specific changes can be seen in the blue font part of the paper.

b. Moreover, you claimed that “the higher the harmonic order, the less the harmonic current content, so it is usually only necessary to consider within 25 times” and only focused on 5th, 7th and 11th harmonic suppression. But, actually, the big issue in grid-connected converters is the switching harmonics, which you did not consider in the paper. You need to reconsider it and compare the ability of the proposed active power filter with the passive power filters. Furthermore, you are using an LCL filter, which is enough to suppress the harmonics is designed properly. So, what is the benefit of the proposed technique?

Response: Thanks for your sincere suggestions. The papers recommended by the reviewer are very good and valuable, so we have added them to the references. We have modified the introduction and made a more comprehensive discussion of passive filters. Specific changes can be seen in the blue font part of the paper.

Response: Thank you for your sincere suggestions and questions. We are very sorry. Here is our statement error. The correct statement should be that the harmonics provided by the nonlinear load used in this experiment are mainly within 20th, of which 5th, 7th, and 11th harmonic content is more. You mentioned that the big problem of the grid-connected converter is the switching harmonics. We very much agree with your opinion. For the switching harmonics, we have used the LCL filter to filter out. The high-frequency harmonics can be suppressed by properly designing the LCL filter. The technology we proposed is used to filter out the low-order harmonics in the power grid. We have used the blue font to mark the changes in the paper.

Comment (4): How did you select the parameters values?

Response: This is really a good question. We add parameter analysis and tuning methods in this paper, and quote a lot of references to facilitate readers to conduct in-depth research. Specific changes can be seen in the blue and red font part of the paper, such as lines 222-230, lines 254-265, lines 272-274, and lines 374-395 in the text.

Comment (5): Figures 17(b), 18(b), 17(d), and 18(d) were not explained and mentioned in the text.

Response: Thank you for your sincere suggestions. We have thoroughly reviewed all the images in the article and have modified to those that were not mentioned or explained in the text.

Comment (6): The HIL advantages should be explained like those in new references High-frequency harmonics suppression in high-speed railway through magnetic integrated LLCL filter and Magnetic integrated double-trap filter utilizing the mutual inductance for reducing current harmonics in high-speed railway traction inverters. Otherwise, the real experiments are required.

7. The conclusion should be point wise.

Response: Thanks for your sincere suggestions. The article you recommended is particularly excellent. We have made the necessary modifications based on the article you recommended and highlighted the text in blue.

Comment (7): The conclusion should be point wise.

Response: Thanks for your sincere suggestions. The authors have revised the “Conclusion”.

Comment (8): In the technical writing, it is not true to write “we, I, our, etc.”. Moreover, you are not only one author, so, it is not true to write “I, my, etc.”. In addition, it is not preferred to thank your institutions. They, of course, help you achieve your research because it is their duties.

Response: Thanks for your sincere suggestions. The authors have revised the “Acknowledgments”.

Comment (9): Many of the references are old. A state-of-the-art (SotA) review is preferred to show the real advantages of the proposed method against the recent literatures. All the references have no DOI. The format of references needs to be unified. Moreover, references need to be written in a standardized way. Many references are conferences papers.

Response: Thanks for your sincere suggestions. We have updated the references, removed the old references and most of the conference papers, and we have selected some of the latest references for discussion in the introduction. All references have been added DOI and modified according to the PLoS ONE header format. We have marked the specific modifications with different color fonts. The blue and red highlights in the introduction part, and the red highlights in the references.

Reviewer 2:

Comment (1): The proposed method, though effective, appears complex, and the manuscript could benefit from a discussion on the feasibility and limitations of implementing this strategy in real-world scenarios.

Response: Thanks for your sincere suggestions. In Conclusion, We discuss the feasibility and limitations of this strategy in real-world scenarios. In this paper, We have used green fonts for highlighting.

Comment (2): While the paper compares the proposed method to traditional PI control, it lacks comparisons with more recent or alternative advanced harmonic suppression techniques.

Response: Thanks for your sincere suggestions. We add LADRC control strategy as a comparison object in the Results and discussion section, and we have used the green font to highlight the modified part.

Comment (3): The LADRC and QPR controllers involve multiple parameters. Providing guidelines or a systematic approach for parameter tuning would make the methodology more accessible.

Response: Thanks for your sincere suggestions. We add parameter analysis and tuning methods in this paper, and quote some references to facilitate readers to conduct in-depth research. Specific changes can be seen in the blue and red font part of the paper, such as lines 222-230, lines 254-265, lines 272-274, and lines 374-395 in the text.

---

## [Decision Letter · Decision Letter 1]

20 Jan 2025

Active power filter control strategy based on repetitive quasi-proportional resonant control with linear active disturbance rejection control

PONE-D-24-40676R1

Dear Dr. Zhu,

We’re pleased to inform you that your manuscript has been judged scientifically suitable for publication and will be formally accepted for publication once it meets all outstanding technical requirements.

Kind regards,

Akhtar Rasool, Ph.D.

Academic Editor

PLOS ONE

Additional Editor Comments (optional):

The reviewers have mentioned a few corrections to be made in the manuscript. However, they have suggested the revision has made acceptable improvements in the article. Thus, this is considered for acceptance, subject to the minor corrections.

Reviewers' comments:

Reviewer's Responses to Questions

**Comments to the Author**

1. If the authors have adequately addressed your comments raised in a previous round of review and you feel that this manuscript is now acceptable for publication, you may indicate that here to bypass the “Comments to the Author” section, enter your conflict of interest statement in the “Confidential to Editor” section, and submit your "Accept" recommendation.

Reviewer #1: All comments have been addressed

Reviewer #3: All comments have been addressed

2. Is the manuscript technically sound, and do the data support the conclusions?

Reviewer #1: Yes

Reviewer #3: Yes

3. Has the statistical analysis been performed appropriately and rigorously? 

Reviewer #1: N/A

Reviewer #3: Yes

4. Have the authors made all data underlying the findings in their manuscript fully available?

Reviewer #1: Yes

Reviewer #3: Yes

5. Is the manuscript presented in an intelligible fashion and written in standard English?

Reviewer #1: Yes

Reviewer #3: Yes

6. Review Comments to the Author

Reviewer #1: Thank you for the revisions. The revised manuscript can be accepted for publication. ‎Congratulations.

I only have a minor notice about Figures 17(A-C) and 19(D-F), which can be considered in the final version. These figures were not explained and mentioned in the text. They need to be explained and mentioned in the text. The paper is not required to be sent again to the reviewer. You just consider this note before publishing the paper. Congratulations‎ again.

Reviewer #3: This is a good manuscript and the revision version has addressed all comments suggested by the reviewer.

However, a minor correction regarding;

Is 'Parameter Turning' correct? I believe it should be 'Parameter Tuning'

The images of waveform are not clear. The numbers at x and y axis couldn't be read clearly.

7. PLOS authors have the option to publish the peer review history of their article (what does this mean?). If published, this will include your full peer review and any attached files.

Reviewer #1: No

Reviewer #3: No

---

## [Editor Report · Acceptance letter]

PONE-D-24-40676R1

PLOS ONE

Dear Dr. Zhu,

I'm pleased to inform you that your manuscript has been deemed suitable for publication in PLOS ONE. Congratulations! Your manuscript is now being handed over to our production team.

Kind regards,

on behalf of

Dr. Akhtar Rasool

Academic Editor

PLOS ONE